# Key determinants of the dual clamp/activator function of Complexin

**Mazen Makke, Alejandro Pastor-Ruiz, Antonio Yarzagaray, Surya Gaya, Michelle Zimmer, Walentina Frisch, Dieter Bruns***

Center for Integrative Physiology and Molecular Medicine, School of Medicine, University of Saarland, Homburg, Germany

## eLife assessment

This **important** work shows **compelling** data that significantly advances our understanding of the regulation of neurotransmitter and hormone secretion by exploring the mechanisms of how the protein complexin 2 (Cplx2) interacts with the calcium sensor synaptotagmin. The function of mammalian Cplx2 is studied using chromaffin cells derived from Cplx2 knock out mice as a system to overexpress and functionally characterize mutant Cplx2 forms and the interaction between Cplx2 and synaptotagmin. The authors identify structural requirements within the protein for Cplx's dual role in preventing premature vesicle exocytosis and enhancing evoked exocytosis. The findings are of broad interest to neuroscientists and cell biologists.

*For correspondence:
dieter.bruns@uks.eu

Competing interest: The authors declare that no competing interests exist.

## Abstract

Complexin determines magnitude and kinetics of synchronized secretion, but the underlying molecular mechanisms remained unclear. Here, we show that the hydrophobic face of the amphipathic helix at the C-terminus of Complexin II (CpxII, amino acids 115–134) binds to fusion-promoting SNARE proteins, prevents premature secretion, and allows vesicles to accumulate in a release-ready state in mouse chromaffin cells. Specifically, we demonstrate that an unrelated amphipathic helix functionally substitutes for the C-terminal domain (CTD) of CpxII and that amino acid substitutions on the hydrophobic side compromise the arrest of the pre-fusion intermediate. To facilitate synchronous vesicle fusion, the N-terminal domain (NTD) of CpxII (amino acids 1–27) specifically cooperates with synaptotagmin I (SytI), but not with synaptotagmin VII. Expression of CpxII rescues the slow release kinetics of the $Ca^{2+}$-binding mutant Syt I R233Q, whereas the N-terminally truncated variant of CpxII further delays it. These results indicate that the CpxII NTD regulates mechanisms which are governed by the forward rate of $Ca^{2+}$ binding to Syt I. Overall, our results shed new light on key molecular properties of CpxII that hinder premature exocytosis and accelerate synchronous exocytosis.

## Introduction

The accumulation of vesicles in a release ready state is an essential property to meet the speed requirements of synchronized SNARE-mediated exocytosis (*Wickner and Schekman, 2008*; *Südhof and Rothman, 2009*; *Jahn and Fasshauer, 2012*). One of the key questions is how the assembly of SNARE complexes is coordinately paused to allow rapid synchronous neurotransmitter release upon intracellular $Ca^{2+}$ increase. Mechanistically, the required metastable pre-fusion intermediate could be maintained by the action of a SNARE-interacting protein acting as a transient fusion 'clamp'. However, the existence and identity of the putative 'fusion clamp' factor has long been debated (for review, see *Mohrmann et al., 2015*; *Trimbuch and Rosenmund, 2016*; *Brunger et al., 2019*). Complexin (Cpx) is a small cytosolic SNARE regulatory protein that prevents premature and asynchronous vesicle

fusion, and accelerates $Ca^{2+}$-triggered synchronized exocytosis (*Xue et al., 2007*; *Dhara et al., 2014*). Several studies have suggested an inhibitory effect of the accessory α-helix of CpxII, although various mechanisms have been proposed, including direct binding to SNAREs or other proteins (*Giraudo et al., 2009*; *Lu et al., 2010*; *Yang et al., 2010*; *Krishnakumar et al., 2011*; *Kümmel et al., 2011*; *Bykhovskaia et al., 2013*; *Cho et al., 2014*; *Malsam et al., 2020*), electrostatic membrane interactions (*Trimbuch et al., 2014*), or stabilization of the secondary structure of the central helix (*Radoff et al., 2014*). The C-terminal region of Cpx also prevents spontaneous fusion in neurons (*Cho et al., 2010*; *Martin et al., 2011*; *Kaeser-Woo et al., 2012*) and premature secretion in neuroendocrine cells (*Dhara et al., 2014*). In a previous study, we presented evidence that the C-terminus of CpxII may compete with the SNAP-25 SN1 region for binding to SNARE partners, thus disrupting the progressive SNARE complex formation prior to the actual $Ca^{2+}$ stimulus (*Makke et al., 2018*). Furthermore, unlike the accessory α-helix (*Radoff et al., 2014*), the C-terminal domain (CTD) of CpxII cannot be functionally replaced by an unrelated helical sequence (*Makke et al., 2018*), suggesting that specific structural features or key residues within the CTD are required for the clamping function of Cpx.

Independent of the clamping function of Cpx, knockout (ko) and knockdown studies of Cpx have shown, even in the absence of premature spontaneous fusion, a significant reduction in evoked release, indicating a supporting role of Cpx in synchronous neurotransmitter release (*Reim et al., 2001*; *Xue et al., 2007*; *Cai et al., 2008*; *Maximov et al., 2009*; *Strenzke et al., 2009*; *Xue et al., 2009*; *Cho et al., 2010*; *Hobson et al., 2011*; *Martin et al., 2011*; *Jorquera et al., 2012*; *Lin et al., 2013*; *Yang et al., 2013*; *Dhara et al., 2014*; *Kurokawa et al., 2015*; *López-Murcia et al., 2019*). Furthermore, there is increasing evidence that the major fusion-promoting function of complexin in vertebrates is mediated by its very N-terminus, an action that is mechanistically independent and even separable from the clamping function of complexin (*Dhara et al., 2014*). Yet, no consensus has been reached on the mechanism by which the N-terminus facilitates secretion, including changes in the $Ca^{2+}$ affinity of the release machinery (*Dhara et al., 2014*), a stabilizing role for SNARE proteins (*Xue et al., 2010*) and/or direct membrane binding (*Lai et al., 2016*).

Using viral expression of CpxII or its mutants in chromaffin cells, we identify the amphipathic character of the helical domain at the end of CpxII's CTD as the critical molecular determinant for the protein's inhibitory function. Furthermore, our experiments indicate that the cluster of glutamate residues upstream of the amphipathic α-helix may assist synaptotagmin I (SytI) in releasing Cpx's clamp to trigger fast exocytosis. Moreover, our results show that the N-terminal domain (NTD) of CpxII specifically modulates SytI- but not synaptotagmin VII (SytVII)-mediated exocytosis. In this way, the N-terminus of complexin speeds up exocytosis triggering that is controlled by the forward rate of $Ca^{2+}$ binding to the calcium sensor SytI (*Voets et al., 2001*; *Sørensen et al., 2003*). Overall, our results provide new insights into the molecular determinants required for the inhibitory function of CpxII and elucidate the mechanisms facilitating fusion through the CpxII NTD.

## Results

### CpxII inhibits premature exocytosis via the amphipathic α-helix of the CTD

CpxII is the only Cpx isoform expressed in mouse chromaffin cells (*Cai et al., 2008*). To elucidate the mode of action of CpxII in controlling Ca-triggered exocytosis, synchronous secretion of isolated mouse chromaffin cells was recorded by membrane capacitance measurements (CM) in response to photolytic uncaging of intracellular $Ca^{2+}$ [Ca]i using the caged $Ca^{2+}$-compound nitrophyenyl-EGTA. Changes in [Ca]i were monitored using a combination of $Ca^{2+}$ indicators (fura-2 and furaptra). The results show that CpxII ko strongly reduced $Ca^{2+}$-triggered synchronized vesicle fusion (*Figure 1B*), which was accompanied by a profound increase in asynchronous exocytosis during the loading phase with NP-EGTA (free [Ca]i~700 nM, *Figure 1E and F*) to allow for $Ca^{2+}$-dependent priming of chromaffin granules. Expression of the wild-type (WT) protein (CpxII ko+CpxII) fully restored the typical flash-evoked response with a prominent exocytotic burst (EB, *Figure 1B*) and prevented any granule loss by premature exocytosis (*Figure 1E and F*). Thus, CpxII hinders premature vesicle exocytosis, that would otherwise outpace phasic secretion, agreeing with our previous observations (*Dhara et al., 2014*; *Makke et al., 2018*). Both components of the EB, the readily releasable pool (RRP) and the slowly releasable pool (SRP), were similarly affected, without altering the sustained rate (SR) of secretion

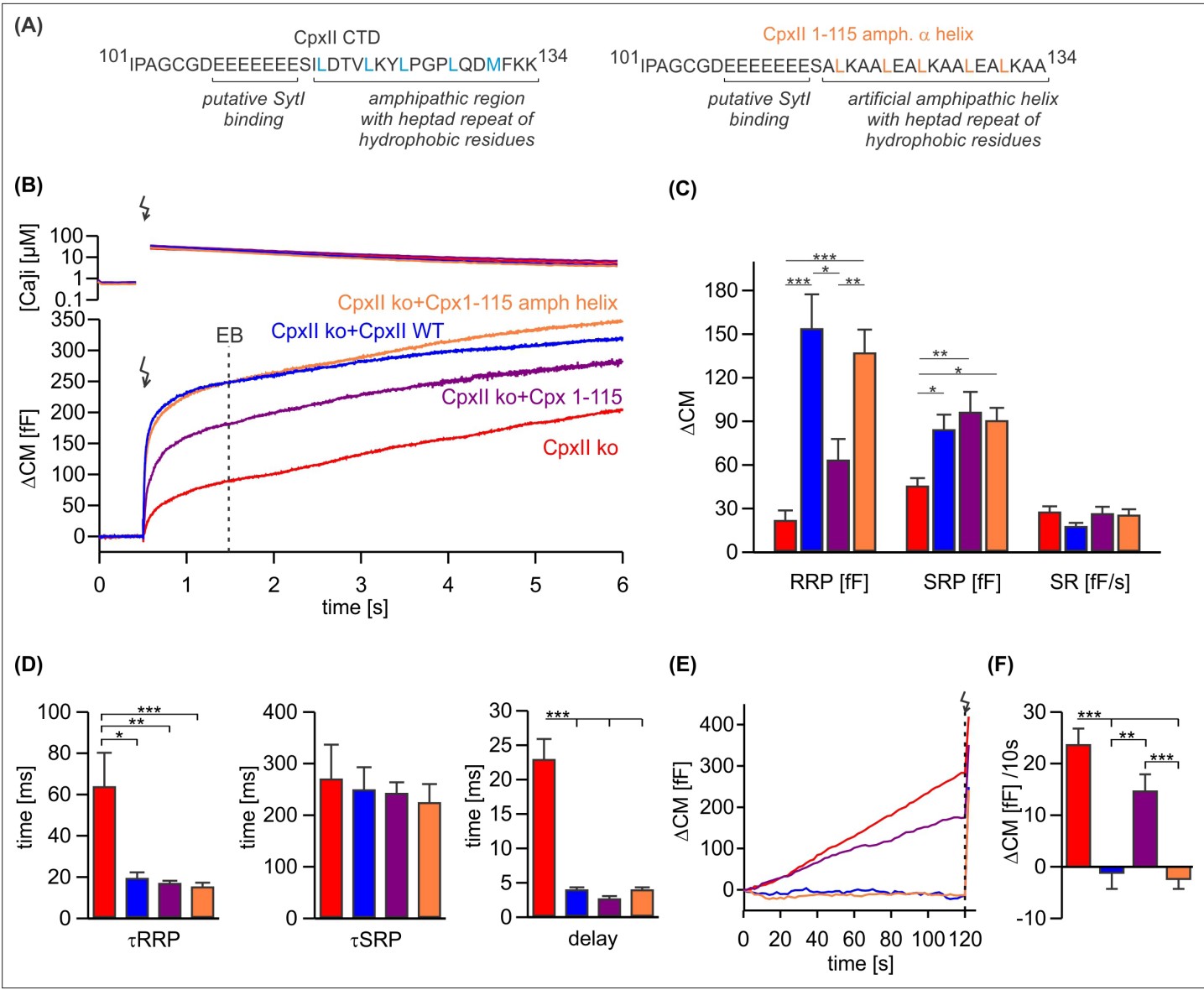

**Figure 1.** An unrelated amphipathic helix functionally substitutes for the last 19 amino acids of CpxII C-terminal domain (CTD). (**A**) Primary sequence of CpxII's CTD (left) and the mutant CpxII 1–115 amphipathic α-helix (right). (**B**) Mean [Ca]i levels (upper panel) and the corresponding membrane capacitance response (lower panel) from CpxII knockout (ko) cells (red n=15), and those expressing either wild-type (WT) CpxII (blue n=15), the CpxII 1–115 mutant (purple n=14), or the CpxII 1–115 amphipathic helix mutant (orange n=16). (**C**) Amplitudes of the readily releasable pool (RRP), the slowly releasable pool (SRP), and the rate of sustained release (SR). CpxII 1–115 amphipathic helix mutant fully supports exocytosis like the WT protein. (**D**) The time constants for the exocytotic burst (EB) components ($\tau$ RRP and $\tau$ SRP), and the exocytotic delay are rescued by the CpxII mutants like the WT protein. (**E**) Mean premature secretion of the tested groups at similar submicromolar [Ca]i before triggering the flash response shown in (**B**). (**F**) The rate of tonic exocytosis (determined at similar [Ca$^{2+}$]i: in nM, CpxII ko: 789±120; CpxII ko+CpxII: 777±57; CpxII ko+CpxII 1–115: 739±84; CpxII ko+CpxII1-115 amph helix: 796±64) is significantly reduced with CpxII and CpxII 1–115 amph helix mutant but not with the truncated CpxII 1–115 mutant. ANOVA or Kruskal-Wallis followed by corresponding post hoc test. **p<0.01; ***p<0.001. Error bars indicate mean ± SEM.

The online version of this article includes the following source data for figure 1:

**Source data 1.** Excel file containing quantitative data.

(*Figure 1C*). Furthermore, loss of CpxII conferred a slower time constant of the RRP response and a longer secretory delay (*Figure 1D*). Overall, CpxII not only hinders premature secretion but also accelerates exocytosis timing. The inhibitory function of CpxII resides within the last 34 amino acids of its CTD (*Makke et al., 2018*), which is characterized by an amphipathic helix and an upstream located cluster of glutamate residues (*Figure 1A*). Yet, the precise molecular mechanism of its clamp action

remained unclear. Truncation of the amphipathic helix (CpxII 1–115) strongly reduced the RRP component of the EB (*Figure 1B*) and elevated premature secretion (*Figure 1E and F*) when compared with the WT protein. Surprisingly, the CpxII 1–115 amp helix mutant, in which the last 19 amino acids were replaced with an unrelated amphipathic helix (*Figure 1A*), fully restored the flash-evoked response and the ability of CpxII to prevent premature secretion (*Figure 1B–F*). As the CpxII NTD determines the stimulus-secretion coupling (*Dhara et al., 2014*; *Makke et al., 2018*), both the RRP secretion kinetics and the secretory delay was fully rescued by the mutant variants (*Figure 1D*). Therefore, the amphipathic helix of CpxII's CTD is a key determinant for the protein to prevent premature secretion.

## The hydrophobic face of C-terminal amphipathic helix of CpxII is crucial for its inhibitory function

To investigate whether specific amino acids within the CpxII CTD amphipathic α-helix are essential for its inhibitory function, we introduced point mutations on either the hydrophobic or hydrophilic face of the helix and tested their impact on regulated exocytosis. Perturbation of the hydrophobic face of the CpxII CTD amphipathic helix by replacing the two hydrophobic leucine amino acids L124 and L128 with charged glutamate residues (CpxII L124E-L128E) mimicked the phenotype of the truncated CpxII 1–115 variant (*Figure 2A–C* compare with *Figure 1*). The CpxII L124E-L128E mutant only partially restored the EB (*Figure 2B*) and largely failed to clamp premature vesicle exocytosis (*Figure 2C*). In contrast, replacing the same leucine residues with hydrophobic tryptophan amino acids (CpxII L124W-L128W) fully rescued the EB size (*Figure 2E*), and prevented premature exocytosis like WT CpxII (*Figure 2F*). No changes in the kinetics of synchronized exocytosis were detected for either mutant variant (*Figure 2—figure supplement 1*). In the same line, mutating L117 and L121 to either glutamate (CpxII L117E-L121E) or tryptophan (CpxII L117W-L121W) confirmed the result that preservation of the hydrophobic face of the amphipathic helix is a prerequisite for arresting premature fusion (*Figure 2—figure supplement 2*). We have previously pointed out that the CpxII CTD shares a high degree of structural similarity with the C-terminal half of the SNAP25-SN1 domain and showed that corresponding CpxII-SNAP25-SN1 chimeras (residues 44–77) fully restore the function in CpxII-deficient cell (*Makke et al., 2018*). In addition to the identical pattern of hydrophobic amino acids (heptad repeat), the sequence comparison between CpxII CTD and SNAP-25-SN1 reveals similar or even identical amino acids on the polar side of the amphipathic helix (*Figure 2—figure supplement 3A*). As a first approach, we replaced these amino acids with alanine residues (D118A, Q129A, D130A, K133A) to test their functional significance. However, none of these point mutants showed an obvious phenotype, neither in the synchronous nor in the asynchronous secretion response (*Figure 2—figure supplement 3*). Similarly, the double point mutant D118K-D130K, with an intended charge reversal, had no serious functional consequences compared to the response with the WT protein (*Figure 2—figure supplement 4*). These results agree with our observation that even an unrelated amphipathic helix can functionally replace the CTD of CpxII. Collectively, they indicate that preserving the hydrophobic face of the amphipathic helix at the end of the CTD of CpxII is necessary to prevent premature exocytosis.

## The ability of the CpxII CTD to rescue secretion parallels its interaction with SNAREs and SytI

Since the CpxII CTD transiently obstructs the assembly of SDS-resistant SNARE complexes (*Makke et al., 2018*), it is possible that the CpxII C-terminus with its SN1 mimetic region competes with the SN1 motif (membrane-proximal layers) of SNAP25 for binding to its cognate SNARE partners, thereby arresting exocytosis. To probe whether CpxII CTD and its mutant variants display altered SNARE interactions, immobilized GST-CpxII CTD or its mutants were incubated with detergent extract from WT mouse brain. Interaction partners were eluted by thrombin cleavage (STH-fraction). While no binding to GST alone was observed, the GST-CpxII CTD pulled down Synaptotagmin I (SytI), Syntaxin 1A (Syx1), and traces of SynaptobrevinII (SybII) (*Figure 3A*). CpxII CTD-SNARE interactions and binding to SytI were abolished for the 'loss of clamp' mutant CpxII CTD L124E-L128E (*Figure 3A*). In contrast, they were preserved or even enhanced for the CpxII CTD L124W-L128W mutant protein with undiminished 'clamp' function. Furthermore, the CpxII 1–115 amphipathic helix mutant efficiently co-precipitated SytI, Syx1, and SybII (*Figure 3B*). Accompanying immunofluorescence analyses reveal similar expression levels for CpxII and its mutant variants (*Figure 3—figure supplement 1*) and show that

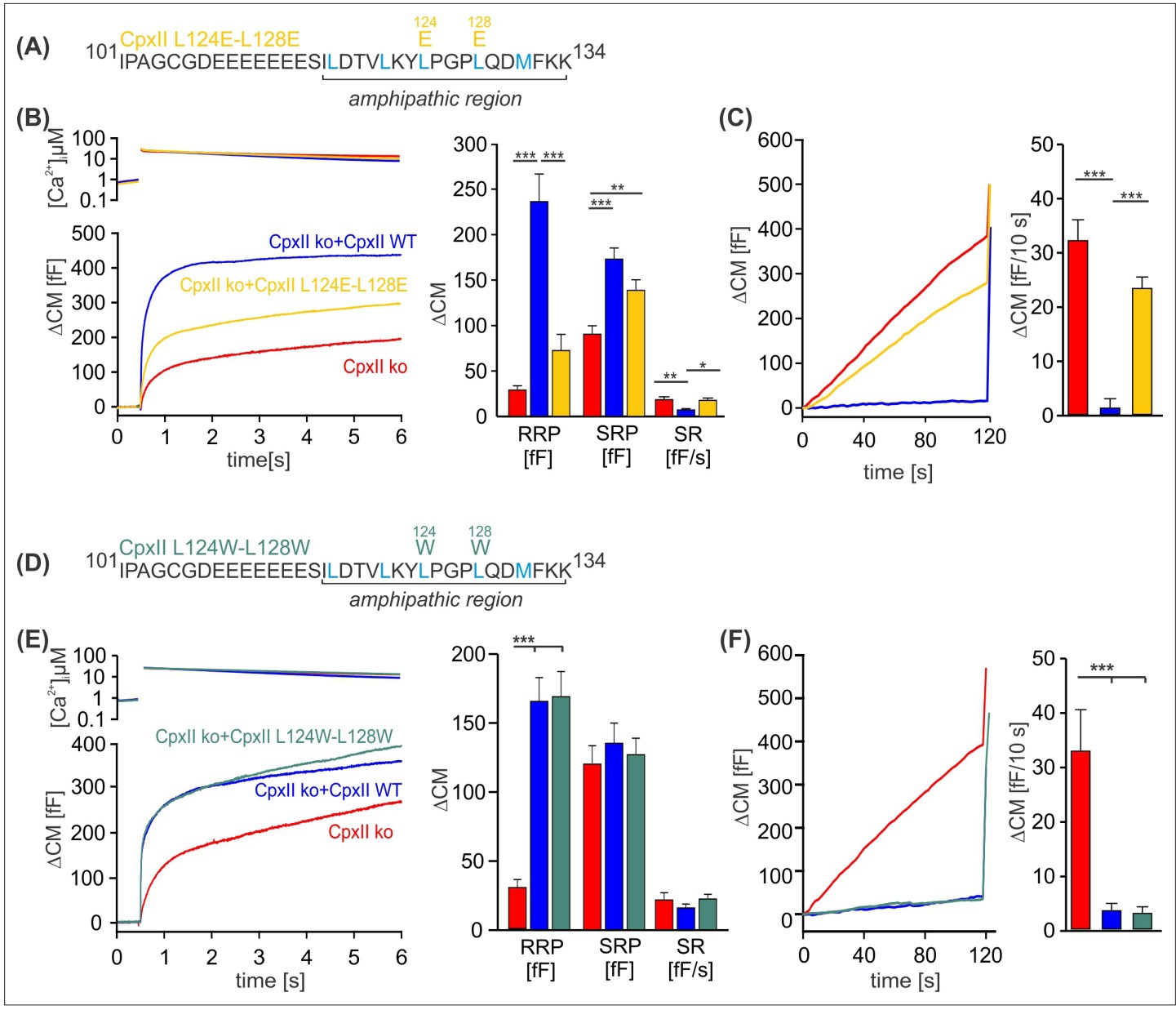

**Figure 2.** Disrupting the hydrophobic face of the amphipathic helix impairs the ability of the CpxII C-terminal domain (CTD) to prevent premature release. (**A, D**) Primary sequences depicting the point mutations L124E-L128E (**A**) and L124W-L128W (**D**) within the amphipathic region of CpxII CTD. (**B, E**) Mean [Ca]i levels (upper panel) and the corresponding ΔCM (lower panel) from CpxII knockout (ko) cells (red n=25), and those expressing CpxII wild-type (WT) (blue n=25), or the mutants CpxII L124E-L128E (yellow n=26) in (**B**), and CpxII ko cells (red n=18) or those expressing the CpxII WT protein (blue, n=15) or CpxII LL-WW (dark green, n=12) in (**D**). Mean amplitudes of the readily releasable pool (RRP), slowly releasable pool (SRP), and sustained rate (SR) show a significant reduction in the size of the exocytotic burst (EB) by the CpxII L124E-L128E mutant, whereas CpxII L124W-L128W rescued the synchronized exocytosis like CpxII WT. (**C, F**) Average tonic exocytosis at similar submicromolar [Ca]i levels. Note that the mutant CpxII L124W-L128W fully clamps premature secretion like the CpxII WT protein, but the CpxII L124E-L128E mutant fails to do so. ANOVA or Kruskal-Wallis followed by corresponding post hoc test. *p<0.05; **p<0.01; ***p<0.001. Error bars indicate mean ± SEM.

The online version of this article includes the following source data and figure supplement(s) for figure 2:

**Source data 1.** Excel file containing quantitative data.

**Figure supplement 1.** The CpxII C-terminal domain (CTD) mutants CpxII L124E-L128E and CpxII L124W-L128W rescue the kinetics of synchronized exocytosis.

**Figure supplement 1—source data 1.** Excel file containing quantitative data.

**Figure supplement 2.** The hydrophobic face of CpxII's amphipathic helix is essential to suppress premature release.

*Figure 2 continued on next page*

*Figure 2 continued*

**Figure supplement 2—source data 1.** Excel file containing quantitative data.

**Figure supplement 3.** Alanine substitutions on the polar face of CpxII C-terminal domain (CTD) have no impact on its clamping ability.

**Figure supplement 3—source data 1.** Excel file containing quantitative data.

**Figure supplement 4.** The charge reversal mutation D118K-D130K does not affect the inhibitory function of the CpxII C-terminal domain (CTD).

**Figure supplement 4—source data 1.** Excel file containing quantitative data.

the latter co-localize with SybII like the WT protein, indicating unperturbed sorting to chromaffin granules and lipid binding (*Figure 3—figure supplement 2*). Taken together, the inhibition of premature exocytosis by CpxII CTD and its mutants correlates with their ability to interact with SytI and SNARE proteins, suggesting that these interactions are a prerequisite for CpxII CTD to prevent premature exocytosis.

## The cluster of glutamates in the CpxII CTD speeds up exocytosis timing

Further truncation of the C-terminus between amino acid position 100–115 exacerbates the 'unclamping' phenotype (*Makke et al., 2018*), indicating that this region provides structural elements that contribute to or stabilize the fusion-arresting function of the amphipathic helix. The region upstream to the amphipathic helix is characterized by a cluster of glutamate residues (aa 108–114) which is conserved across the animal kingdom and, according to biochemical experiments, may serve as a putative SytI interaction site (*Tokumaru et al., 2008*). However, the functional role of the glutamate cluster in regulated exocytosis remains to be elucidated. To follow up on this, we replaced the glutamate cluster with corresponding stretch of alanine residues (CpxII E-A). When expressed in CpxII ko chromaffin cells, the mutant CpxII E-A (*Figure 4A*) only supported a reduced synchronized EB (*Figure 4B*) with a specifically reduced RRP (*Figure 4C*). In addition, the remaining EB exhibited slower fusion kinetics for both phases of the synchronous exocytosis, the RRP and the SRP, and an increased stimulus-secretion delay when compared to the secretion response of WT CpxII (*Figure 4D*). Nevertheless, premature exocytosis was effectively suppressed by the CpxII E-A mutant, as was observed for the WT protein (*Figure 4E and F*). As the CpxII E-A mutation has effects on the timing of rapid exocytosis, it may interfere with the efficacy of SytI in triggering vesicle fusion. Notably, no comparable slowing of exocytosis was observed when both protein domains, the glutamate cluster and the amphipathic helix, were truncated, such as with the CpxII 1–100 mutant (*Makke et al., 2018*). Thus, with an intact CTD of CpxII, the glutamate cluster may be required as a contact site for SytI to release the molecular clamp of the downstream amphipathic helix and thus to trigger the rapid secretion response.

## The functional interplay of SytI and CpxII in Ca²⁺-triggered exocytosis

To study the potential interplay of SytI and CpxII in triggering secretion, we comparatively analyzed the phenotypic consequences of single and compound deficiencies of these proteins. Consistent with previous work (*Voets et al., 2001*; *Nagy et al., 2006*; *Dhara et al., 2014*), loss of SytI reduced the EB size, specifically by decreasing the RRP component (*Figure 5A–C*), and prolonged the kinetics of SRP secretion as well as the secretory delay when compared to the WT response (*Figure 5D*). In stark contrast, additional loss of SytI in the absence of CpxII did not further aggravate the phenotype, suggesting that all SytI functions are critically dependent on the presence of CpxII. Furthermore, SytI deficiency did not alter premature secretion when compared to the response of WT cells or with the elevated secretion in the absence of CpxII (*Figure 5E and F*). Collectively, these results demonstrate a clear dependence of SytI functions on CpxII for the synchronous secretion response. At submicromolar [Ca]i, changes in the CpxII expression systematically altered the amount of premature secretion, which inversely correlated with the EB size (*Figure 5—figure supplement 1*). Thus, CpxII inhibits premature secretion that would otherwise outpace synchronous release. A similar relationship between tonic secretion and EB size was found in the absence of SytI, indicating that CpxII hinders premature secretion in a SytI-independent manner (*Figure 5—figure supplement 1*). Genetic loss of *Syt7* also reduced the EB size (*Figure 5G and H*), agreeing with previous work (*Schonn et al., 2008*). Both components of the EB, the RRP and the SRP, were similarly affected (*Figure 5H*). The RRP kinetics were found to be slightly faster when compared with controls (*Figure 5I* (inset), J). Strikingly, deletion

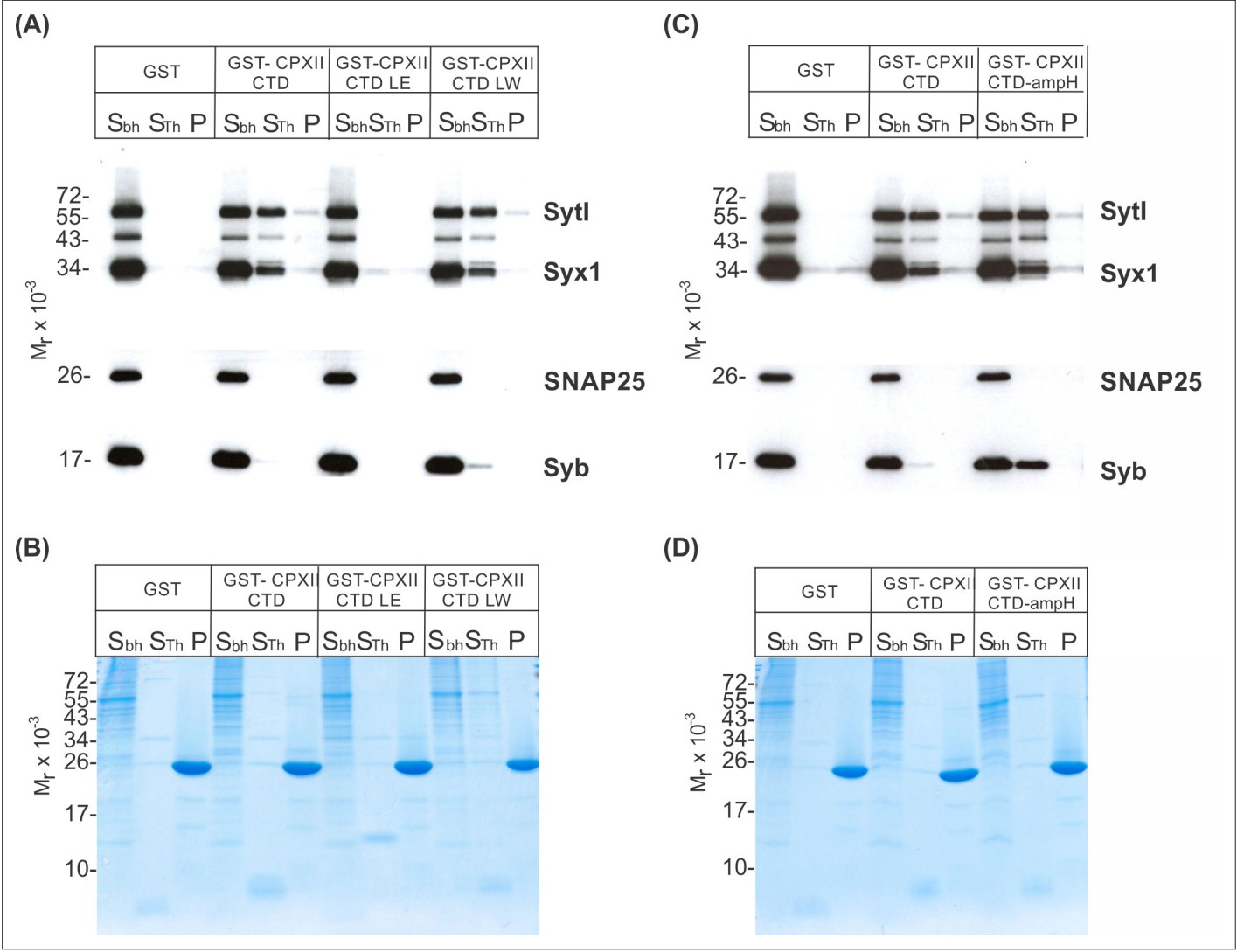

**Figure 3.** CpxII-C-terminal domain (CTD) and its mutant variants differentially interact with SNAREs and SytI. (A, C) The GST-CpxII CTD fusion protein co-precipitates SytI and Syx1a from detergent extract of mouse brain. Substitution of leucine residues with glutamate residues (CpxII-CTD L124E-L128E, LE) abolishes any binding to CpxII CTD (A). Substitution with tryptophan residues (CpxII-CTD L124W-L128W, LW), instead, or replacement of the entire CpxII CTD with an unrelated amphipathic helix (CpxII-CTD-ampH, C) is tolerated. Samples were eluted from the column by thrombin cleavage. Equal volumes of supernatant ($S_{bh}$), the thrombin-eluted ($S_{Th}$), and the non-eluted fraction (P) were analyzed by SDS-PAGE (12% gel) and western blotting with antibodies against the indicated antigens. No binding to GST alone could be detected. (B, D) Corresponding Coomassie gels documenting the integrity of the GST fusion proteins, remaining in the non-eluted fraction (P) after thrombin cleavage.

The online version of this article includes the following source data and figure supplement(s) for figure 3:

**Source data 1.** Western blot and corresponding SDS-PAGE analysis of CpxII-interacting proteins in *Figure 3A and B*.

**Source data 2.** Western blot and corresponding SDS-PAGE analysis of CpxII-interacting proteins in *Figure 3C and D*.

**Figure supplement 1.** Expression analysis of CpxII and its mutants reveals similar levels of protein expression.

**Figure supplement 1—source data 1.** Excel file containing quantitative data of the expression analysis of CpxII and its mutant variants.

**Figure supplement 2.** CpxII and its mutants concentrate at vesicular membranes.

**Figure supplement 2—source data 1.** Excel file containing quantitative data of the co-localization analysis of CpxII and its mutant variants.

of SytVII in the CpxII ko background almost completely eliminated the EB, leaving only a sustained phase of secretion behind (*Figure 5G–I*). The phenotype of the CpxII-SytVII dko is remarkably similar to the compound deficiency of SytI and SytVII (*Schonn et al., 2008*), corroborating the view that CpxII serves as a gatekeeper of SytI function in the Ca²⁺-triggered exocytosis of chromaffin granules. In the same line, CpxII expression in SytVII ko cells enhances exocytosis like in WT cells (*Figure 5—figure*

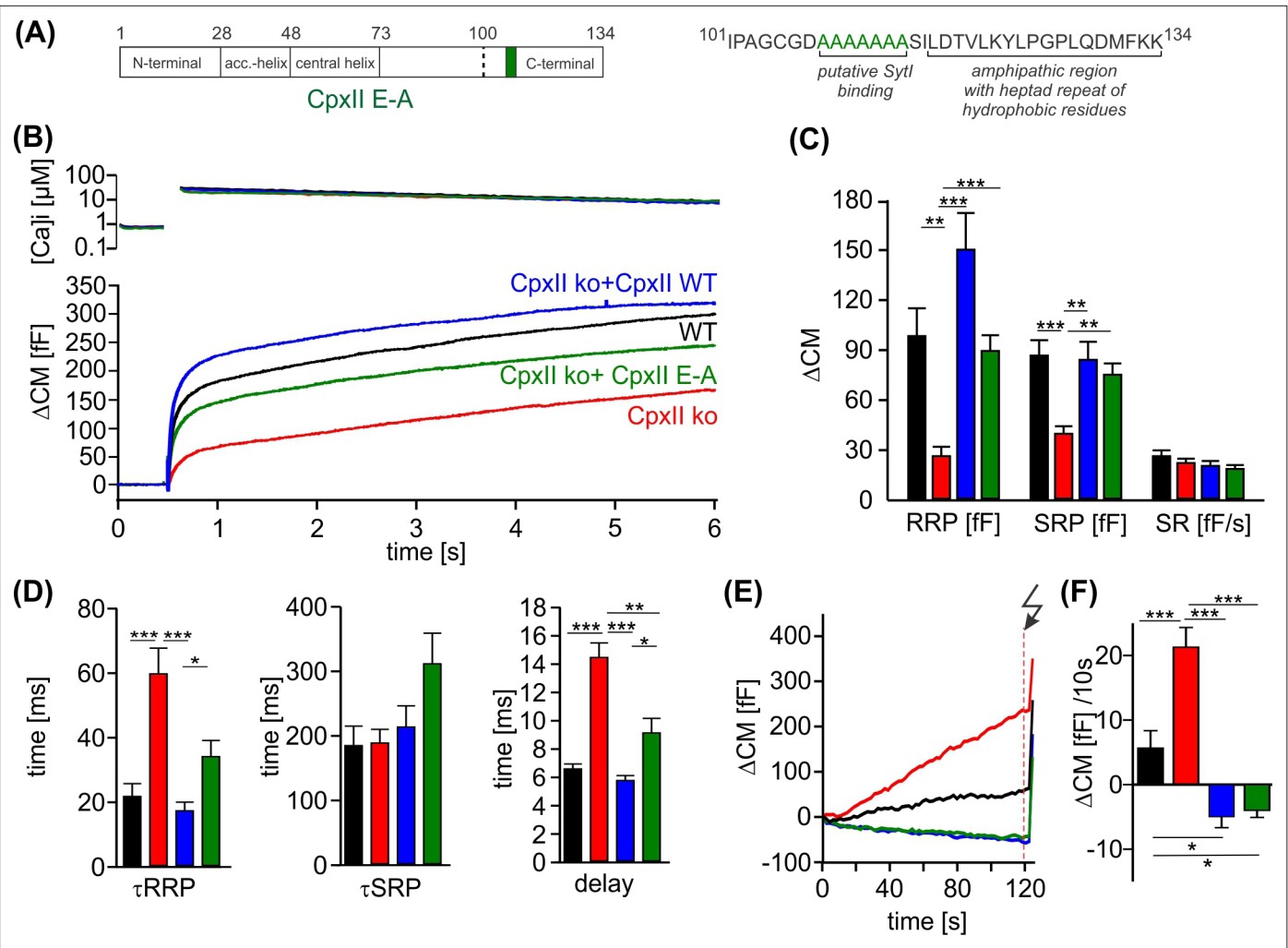

**Figure 4.** The cluster of glutamate residues in CpxII C-terminal domain (CTD) facilitates Ca²⁺-triggered exocytosis. (**A**) Schematic representation of CpxII highlighting the position of the glutamate cluster within the CTD. Primary sequence of CpxII CTD showing the exchange of glutamate into alanine residues (CpxII E-A mutant). (**B**) Mean [Ca]i levels (upper panel) and capacitance measurement (CM) response from wild-type (WT) cells (black, n=18), CpxII knockout (ko) cells (red, n=19), and CpxII ko cells expressing either CpxII WT (blue, n=17) or the mutant CpxII E-A (green, n=24). (**C**) The mutant CpxII E-A only partially restores the fast component (readily releasable pool [RRP]) of synchronized exocytosis. (**D**) The CpxII E-A mutation significantly slows down the speed of the RRP and slowly releasable pool (SRP) and delays the stimulus-secretion coupling. (**E**) Average tonic exocytosis at similar submicromolar [Ca]i levels with the quantification in (**F**) showing that the mutant CpxII E-A effectively prevents premature vesicle loss at submicromolar [Ca]i levels like CpxII WT. ANOVA or Kruskal-Wallis followed by corresponding post hoc test. *p<0.05; **p<0.01; ***p<0.001. Error bars indicate mean ± SEM.

The online version of this article includes the following source data for figure 4:

**Source data 1.** Excel file containing quantitative data.

---

supplement 2). In contrast to the loss of SytI, SytVII deficiency strongly reduced premature secretion when compared with WT cells or on the background of CpxII deficiency (*Figure 6K and L*). Thus, deletion of Syt VII profoundly alters the exocytotic rates at submicromolar [Ca²⁺]i, a notion that agrees with its proposed function as high-affinity Ca²⁺ sensor (*MacDougall et al., 2018*). Overall, the combined set of data reveals a differential interplay between the two Syt isoforms and CpxII. While function of SytI is critically dependent on CpxII, SytVII appears to act independently, most likely in the Ca²⁺-dependent priming reaction (*Tawfik et al., 2021*).

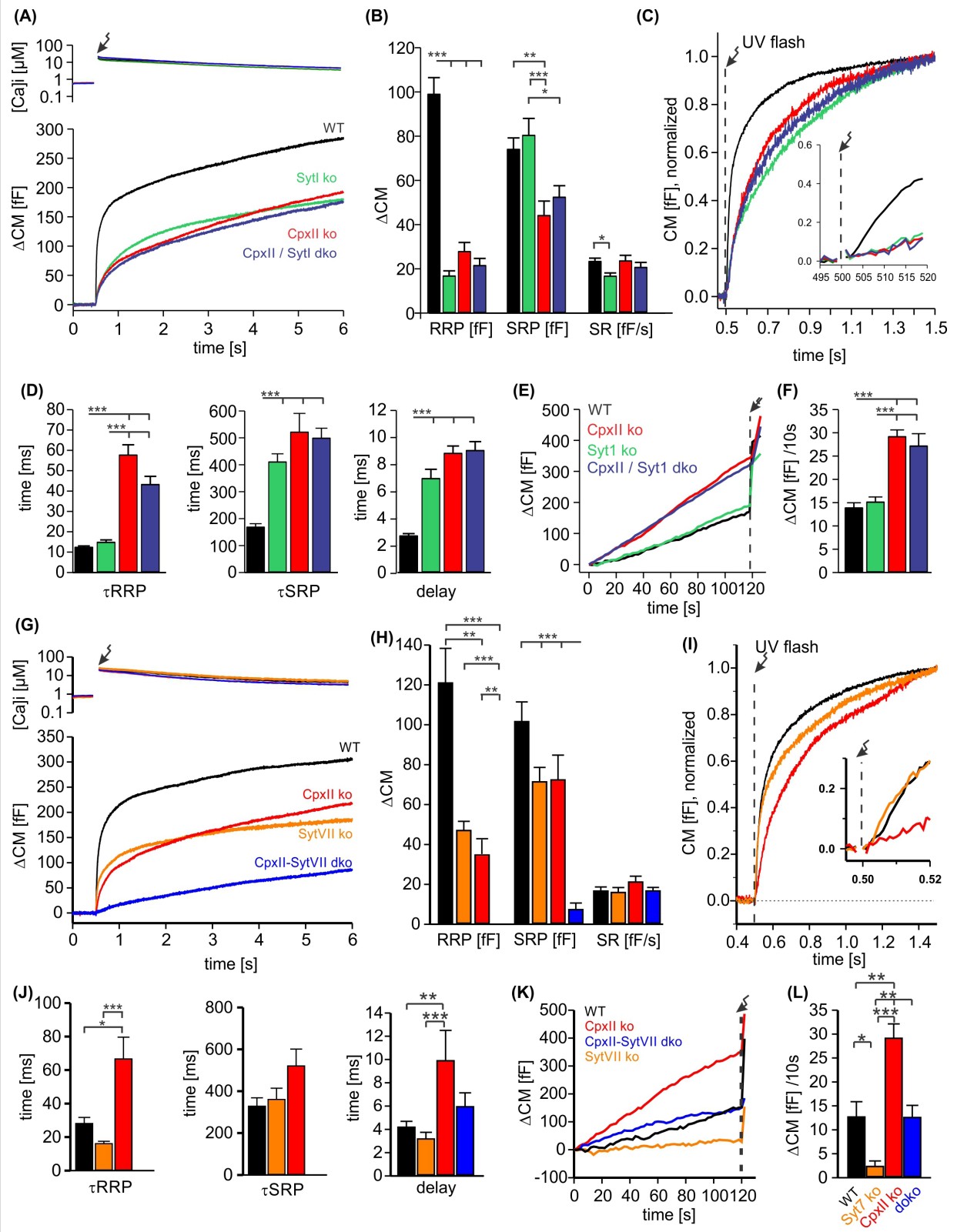

**Figure 5.** CpxII and SytI act interdependently in triggering synchronized exocytosis. (**A**) Mean [Ca]i levels (upper panel) and capacitance measurement (CM) responses from wild-type (WT) cells (black, n=50), SytI knockout (ko) cells (green, n=46), CpxIIko (red, n=35), CpxII/SytI dko (dark blue, n=26). (**B**) Quantification of the readily releasable pool (RRP) and slowly releasable pool (SRP) size shows that SytI deletion in the absence of CpxII does not further aggravate the secretory deficit. (**C**) Normalized CM (as shown in A) scaled to the WT response 1 s after the flash. Inset, extended scaling of normalized

*Figure 5 continued on next page*

*Figure 5 continued*

CM during the first 20 ms after flash (arrow) depicting the delayed onset of secretion. (**D**) Kinetics of RRP and SRP exocytosis and the secretory delay. (**E**) Loss of SytI does not alter premature vesicle fusion. Additional loss of CpxII increases premature exocytosis to the level of CpxII ko (**F**). (**G**) Mean [Ca]i levels (upper panel) and CM responses from WT cells (black, n=17), SytVII ko cells (orange, n=20), CpxII ko (red, n=20), CpxII/SytVII dko (blue, n=24). (**H**) The RRP and SRP sizes are reduced by either CpxII or SytVII single ko, while the combined loss of CpxII and SytVII abolishes the EB. (**I**) Normalized CM (as shown in G) scaled to the WT response 1 s after the flash. Inset, extended scaling of normalized CM during the first 20 ms after flash (arrow). (**J**) Loss of CpxII but not of SytVII slows the time constants of RRP and SRP exocytosis ( $\tau$ RRP, $\tau$ SRP) and prolongs the secretory delay. (**K, L**) SytVII deficiency diminishes the elevated premature exocytosis of CpxII ko cells (SytVII CpxII dko) and reduces it compared to WT cells. ANOVA or Kruskal-Wallis followed by corresponding post hoc test. *p<0.05; **p<0.01; ***p<0.001. Error bars indicate mean ± SEM.

The online version of this article includes the following source data and figure supplement(s) for figure 5:

**Source data 1.** Excel file containing quantitative data.

**Figure supplement 1.** CpxII clamps premature exocytosis independently of SytI.

**Figure supplement 1—source data 1.** Excel file containing quantitative data.

**Figure supplement 2.** CpxII boosts exocytosis in the absence of SytVII.

**Figure supplement 2—source data 1.** Excel file containing quantitative data.

## Acceleration of synchronized exocytosis by the CpxII N-terminus relies on SytI

Phenotypic cues from many model systems as well as in vitro analyses point to a fusion-promoting effect of complexin due to increased Ca²⁺ sensitivity, suggesting the possibility of mechanistic crosstalk between complexin and the Ca²⁺ sensor synaptotagmin (*Reim et al., 2001*; *Tadokoro et al., 2005*; *Huntwork and Littleton, 2007*; *Xue et al., 2007*; *Diao et al., 2012*; *Malsam et al., 2012*; *Dhara et al., 2014*). Indeed, expression of an N-terminally truncated complexin variant (e.g. residues 28–134) in cplxII-deficient chromaffin cells failed to restore normal release rates (*Dhara et al., 2014*) and also reduced the EPSC amplitude of hippocampal neurons in response to the short-lasting action potential evoked Ca²⁺ influx (*Xue et al., 2009*; *Xue et al., 2010*). Given the different apparent calcium-binding rates of SytI and SytVII (*Sugita et al., 2002*; *Pinheiro et al., 2016*), one could speculate that the NTD of Cpx accelerates the kinetics of vesicle fusion by preferentially recruiting SytI as a calcium sensor for rapid exocytosis instead of SytVII. To test this hypothesis, we overexpressed the CpxII ΔN mutant in chromaffin cells lacking either the calcium sensor SytI or SytVII. Overexpression of either WT CpxII or the mutant CpxII ΔN had no effect on the kinetics of exocytosis in cells lacking SytI (*Figure 6A–C*), which demonstrates that CpxII requires SytI to accelerate exocytosis. In contrast, in SytVII ko cells, CpxII ΔN slowed down synchronous secretion and increased the secretory delay (*Figure 6D–F*) as previously reported for overexpression in WT cells (*Dhara et al., 2014*). Furthermore, both, CpxII and CpxII ΔN, hindered premature exocytosis independent of the present Syt isoform, corroborating the view that CpxII hinders SNARE action rather than blocking one of the main Ca²⁺ sensors in chromaffin cells (*Figure 6E, F, K, and L*). Together, these findings support a scenario in which CpxII NTD and SytI are functionally interdependent in the activation of fast synchronous exocytosis. Because the CpxII ΔN mutant slows down secretion in SytVII ko cells as in WT cells (*Dhara et al., 2014*), the CpxII NTD directly regulates SytI rather than acting as a molecular switch between these Syt isoforms.

## CpxII NTD accelerates exocytosis timing by enhancing the apparent Ca²⁺ affinity of the release machinery

Since the promotive role of the NTD of CpxI can only be tested in the presence of SytI, we took advantage of the knockin (ki) SytI R233Q mutant, which carries the substitution mutation in the calcium-binding pocket of the C2A domain and lowers the affinity for Ca²⁺ binding (*Fernández-Chacón et al., 2001*). Intriguingly, the phenotype of the R233Q ki mutant in chromaffin cells is kinetically similar to that of the CpxII ΔN mutant, as it also slows down timing of exocytosis compared to WT cells (*Sørensen et al., 2003*; *Dhara et al., 2014*). Consistent with previous work by *Sørensen et al., 2003*, we found that the SytI R233Q mutation slowed down the rate of exocytosis, prolonged the secretory delay and enhanced the EB size when compared to WT cells (*Figure 7A–D*). Overexpression of the CpxII ΔN mutant in SytI R233Q ki cells, which is expected to outcompete the function of endogenous CpxII in these cells (*Dhara et al., 2014*), further slowed down the rate of synchronized release and restored the EB size to the WT level (*Figure 7C and D*). Remarkably, overexpression of the WT CpxII

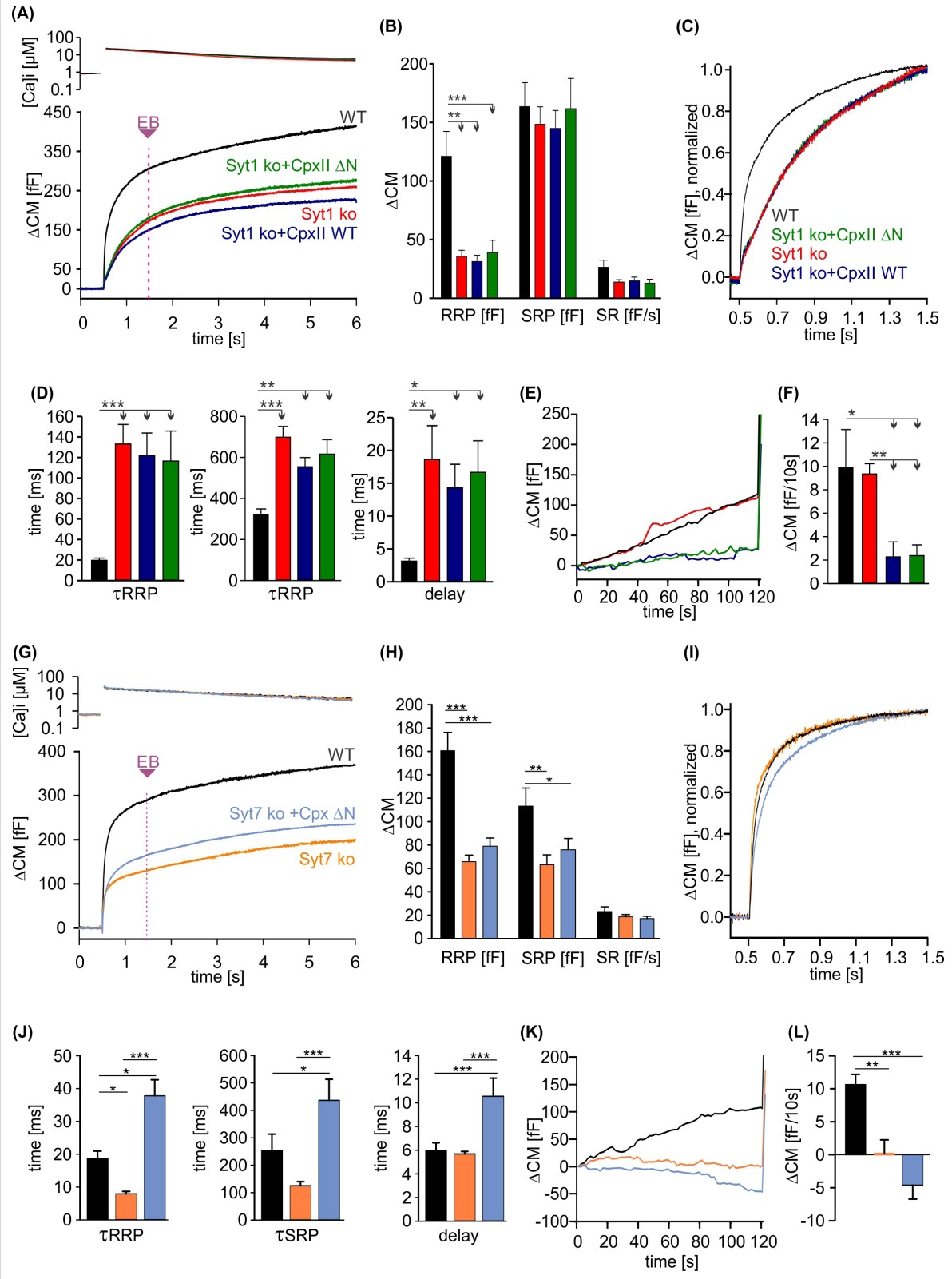

**Figure 6.** CpxII N-terminal domain (NTD) exclusively modulates the kinetics of SytI-mediated exocytosis. (**A**) Mean [Ca]i levels (upper panel) and capacitance measurement (CM) response from wild-type (WT) cells (black, n=13), SytI knockout (ko) cells (red, n=19), and those overexpressing either CpxII WT (dark blue, n=16), or the mutant CpxII ΔN (green, n=20). In the absence of SytI, Cpx ΔN fails to slow down exocytosis timing as observed in SytVII cells (**I**). (**B–D**) Neither CpxII nor CpxII ΔN expression in SytI ko cells alters the size of the pools, the sustained rate (SR) rate (**B**) or the kinetics of

*Figure 6 continued on next page*

*Figure 6 continued*

synchronous exocytosis (C, normalized CM; D, time constants of readily releasable pool [RRP] and slowly releasable pool [SRP] exocytosis and secretory delay). (**E, F**) Both, CpxII and CpxII ΔN expression, hinders asynchronous release in the absence of SytI (**F**). (**G**) Mean [Ca]i levels (upper panel) and CM response of WT cells (black, n=13), SytVII ko cells (orange, n=17), and those expressing CpxII ΔN (light blue, n=29). (**H**) Both the RRP and SRP sizes are reduced in the absence of SytVII. (**I**) Normalized CM responses (of data shown in G) scaled to the WT response 1 s after the flash. Note that CpxII ΔN mutant slows kinetics of release in SytVII ko cells. (**J**) Loss of SytVII speeds up exocytosis timing, which in turn is slowed down by additional expression of CpxII ΔN. (**K, L**) Analysis of premature exocytosis showing reduced premature exocytosis in SytVII ko cells compared to WT. CpxII ΔN suppresses asynchronous release also in the absence of SytVII. ANOVA or Kruskal-Wallis followed by corresponding post hoc test. *p<0.05; **p<0.01; ***p<0.001. Error bars indicate mean ± SEM.

The online version of this article includes the following source data for figure 6:

**Source data 1.** Excel file containing quantitative data.

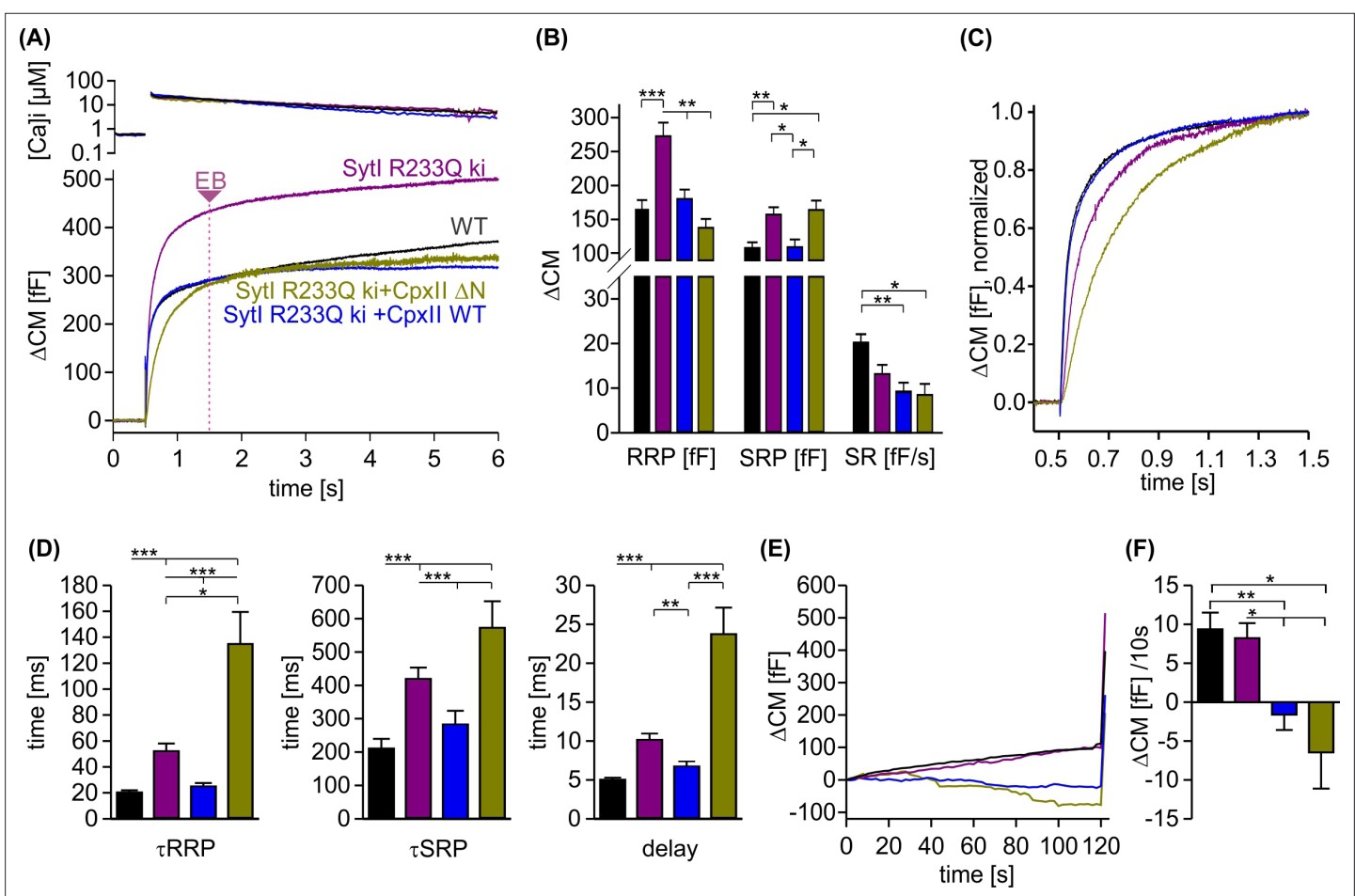

**Figure 7.** CpxII overexpression rescues the slow secretory rates of SytI R233Q knockin (ki) cells. (**A**) Mean [Ca]i levels (upper panel) and capacitance measurement (CM) responses from wild-type (WT) (black, n=35), SytI R233Q ki (violet, n=39), and SytI R233Q ki cells expressing either CpxII WT (blue, n=18) or the mutant CpxII ΔN (olive green, n=11). (**B**) Compared to WT cells, SytI R233Q ki cells show an increased readily releasable pool (RRP) and slowly releasable pool (SRP) size, which are restored by CpxII or CpxII ΔN expression. (**C**) While CpxII speeds up exocytosis timing of SytI R233Q ki cells to the level of WT cells, CpxII ΔN slows it further down. Normalized CM responses (of data shown in A) scaled to the WT response 1 s after the flash. (**D**) CpxII and CpxII ΔN expression oppositely regulate the kinetics of exocytosis (τ RRP, τ SRP) and the secretory delay. (**E, F**) Mean premature exocytosis, determined at similar submicromolar [Ca]i (in nM, SytI R233Q ki:623±31; SytI R233Q ki+CpxII:569±28; SytI R233Q ki+CpxII ΔN:600±26; WT: 582±22), showing that CpxII WT or CpxII ΔN expression effectively hinders premature vesicle secretion. ANOVA or Kruskal-Wallis followed by corresponding post hoc test *p<0.05; **p<0.01; ***p<0.001. Error bars indicate mean ± SEM.

The online version of this article includes the following source data for figure 7:

**Source data 1.** Excel file containing quantitative data.

significantly accelerated the kinetics of synchronous exocytosis in SytI R233Q ki cells, restoring them to the level of WT cells. Furthermore, CpxII WT and its mutant showed an undiminished suppression of premature exocytosis in the SytI R233Q ki cells (*Figure 7E and F*). Thus, CpxII with its NTD is able to compensate for the functional deficiencies of the SytI R233Q mutation. Given that exocytosis timing in chromaffin cells is largely determined by the kinetics of $Ca^{2+}$ binding to SytI (*Voets et al., 2001*; *Sørensen et al., 2003*), the opposite effects of the WT protein and its ΔN mutant on the kinetics of secretion indicate that the NTD of CpxII accelerates mechanisms regulated by the $Ca^{2+}$-binding kinetics of SytI.

## CpxII NTD supports the forward rate of calcium binding to SytI in accelerating exocytosis

To obtain release kinetics over a wider range of [Ca]i, we made use of a calcium ramp protocol, in which calcium was progressively released from the calcium cage by alternating 340 and 380 nm illumination allowing us to combine slow calcium uncaging with ratiometric measurements of [Ca]i (*Sørensen et al., 2003*). Simultaneous CM revealed a sigmoid-shaped capacitance increase during the calcium ramp protocol indicating progressive depletion of the primed vesicle pool and thus providing an estimate of the EB size (*Figure 8A*). To measure fusion kinetics, the slope of the capacitance increase, the remaining pool, and the corresponding exocytosis rates were determined during the $Ca^{2+}$ ramp (*Figure 8B*). Overexpression of CpxII WT protein in SytI R233Q ki cells clearly shifted the half-maximal concentration for pool depletion (P50) to lower [Ca]i, whereas overexpression of the CpxII ΔN mutant increased it (*Figure 8C*). Evidently, WT CpxII rescued the secretion deficit of the Syt R233Q mutant, as observed in the flash experiment (*Figure 7*). For the SytI R233Q ki cells, fusion rates increase as a function of [Ca]i and are shifted with similar slope to either lower [$Ca^{2+}$]i (SytI R233Q+CpxII) or higher [Ca]i (SytI R233Q+CpxII ΔN), indicating unchanged cooperativity of $Ca^{2+}$ binding (*Figure 8D*). The fusion rate-[Ca]i relationships are reasonably well approximated by Hill equations (*Figure 8D*, dashed lines) with similar coefficients (n), but different KD values (SytI R233Q+CpxII, n=2.3, KD = 17.4 μM; SytI R233Q, n=2.03, KD = 38.2 μM; SytI R233Q+CpxII ΔN, n=2.07, KD = 58.13 μM). It should be noted that the rate of exocytosis in the ramp experiment could only be reliably determined at [Ca]i levels below 3, 4, and 6 μM for SytI R233Q+CpxII, SytI R233Q, and SytI R233Q+CpxII ΔN cells, respectively, so that the accuracy of the measurement is not affected by excessive depletion of the pool of primed vesicles. As the $Ca^{2+}$ sensor is expected to be in a 'quasi steady-state' during the slow $Ca^{2+}$ ramp, it is not possible to disentangle whether the observed changes in $Ca^{2+}$ affinity are due to increased forward or decreased backward rates in $Ca^{2+}$ binding to SytI. For the flash-evoked response, instead, the fusion rate is dominated by the forward rate in $Ca^{2+}$ binding. As similar quantitative differences in exocytosis timing were observed under these conditions, the results indicate that the NTD of CpxII and the forward rate of $Ca^{2+}$ binding to SytI act synergistically to accelerate exocytosis timing.

## Discussion

In many secretory systems, Complexin plays a dual role in the regulation of SNARE-mediated vesicle fusion. On the one hand, Cpx inhibits asynchronous exocytosis, thereby enhancing $Ca^{2+}$-triggered synchronized secretion, and on the other hand, it accelerates fusion of primed vesicles upon elevation of intracellular $Ca^{2+}$. Here, we have investigated the molecular determinants required for the CTD of CpxII to clamp premature vesicle exocytosis and delineated molecular mechanisms by which CpxII's NTD accelerates exocytosis timing. Our findings show that the hydrophobic character of the amphipathic helix at the very C-terminus of CpxII is crucial for inhibiting premature vesicle fusion. We also provide evidence that the group of glutamate residues within the CTD of CpxII accelerates exocytosis by lifting the molecular clamp of the downstream amphipathic α-helix, most likely through interactions with SytI. Furthermore, we show that CpxII cooperates exclusively with SytI and modulates the apparent $Ca^{2+}$ affinity of secretion by regulating processes controlled by the forward rate of $Ca^{2+}$ binding to the $Ca^{2+}$ sensor.

## The clamping action of CpxII in $Ca^{2+}$-triggered secretion

The ability to build up a pool of primed vesicles is a central property of many secretory cells. To ensure adequate stimulus-secretion coupling and to prevent premature exocytosis of these vesicles,

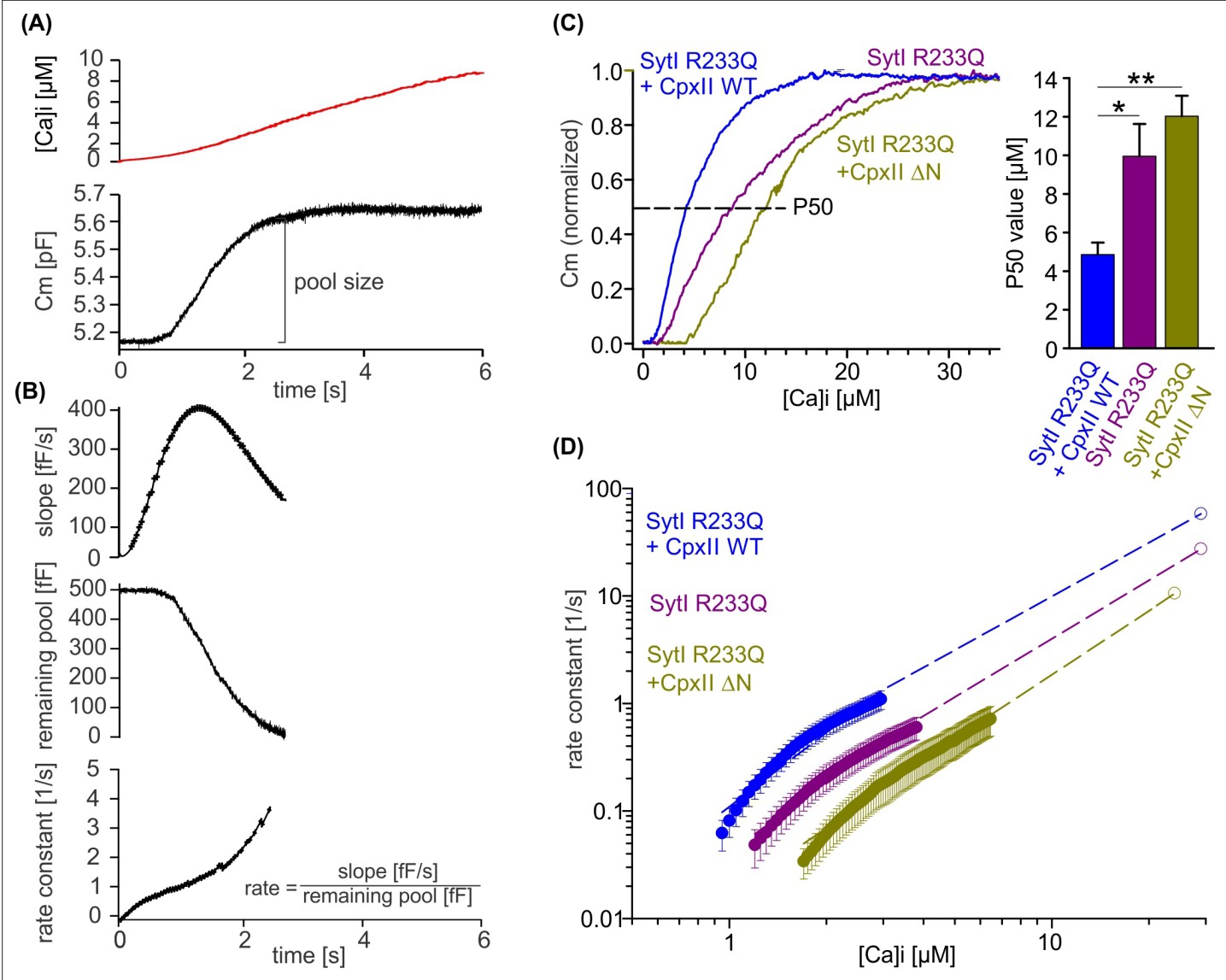

**Figure 8.** CpxII N-terminal domain (NTD) increases the Ca²⁺ affinity of secretion. (**A**) Exemplary capacitance measurement of a chromaffin cell (lower panel) in response to a ramp-like increase in intracellular Ca²⁺ (top panel). (**B**) The slope of the ΔCM (top panel), remaining pool (middle panel), and rate of fusion (lower panel) were determined from the data shown under (**A**) with a temporal interval of 40 ms to match the timing of the ratiometric Ca²⁺ measurement. The rate was determined by dividing the slope [fF/s] by the remaining pool size [fF]. (**C**) Mean profile of pool depletion as the function of [Ca]i (left panel) for SytI R233Q knockin (ki) cells (violet, n=15) and those overexpressing CpxII wild-type (WT) (blue, n=14) or CpxII ΔN (olive green, n=11). Note that expression of CpxII WT lowers [Ca]i for half-maximal pool depletion (**P50**), whereas CpxII ΔN increases it (right panel). (**D**) Double-logarithmic plot of the mean exocytotic rate as the function of [Ca]i for the indicated groups. CpxII overexpression increases the rate of secretion from SytI R233Q ki cells, whereas loss of CpxII NTD further slows it down. Hill equations with similar coefficient, but different KD value, approximate the data (dashed lines). Extrapolation of the rates at low [Ca]i coincides with the readily releasable pool (RRP) kinetics (**Figure 7D**) determined in the flash experiment (hollow circle). ANOVA or Kruskal-Wallis followed by corresponding post hoc test. *p<0.05; ***p<0.001. Error bars indicate mean ± SEM.

The online version of this article includes the following source data for figure 8:

**Source data 1.** Excel file containing quantitative data.

a molecular mechanism must exist that arrests these vesicles in their docked state. Recently, we provided the first evidence that the CTD of CpxII (amino acids 101–134) plays a central role in the molecular clamp of the secretion machinery of chromaffin cells (**Makke et al., 2018**). Thus, it became clear that the CTD of CpxII is essential and rate-limiting to prevent premature fusion and to build up a pool of primed vesicles. Structural similarities between the CTD of CpxII and the C-terminal half of the SNAP25-SN1 domain, as well as the observation that CpxII:SNAP25-SN1 chimeras (C-terminal

half) fully rescued function in CpxII ko cells, suggested that the C-terminus of CpxII may compete with SNAP25-SN1 for binding to the SNARE complex, thereby halting the progressive assembly of the SNARE complex prior to the triggering Ca$^{2+}$ stimulus. However, it remained unclear which structural determinants of the CTD of CpxII mediate its actual inhibitory potential. Our new experiments show that substitution of the CTD with an unrelated amphipathic helix fully restores the function of the CpxII protein (*Figure 1*), suggesting that amphipathicity and most likely preservation of the hydrophobic face is crucial for the inhibitory effect of the CTD. In good agreement, substitution of single hydrophobic leucine residues with charged glutamate residues almost completely abolished the inhibitory phenotype of CTD (*Figure 2*), whereas replacement with equally hydrophobic tryptophan residues was functionally tolerated. Furthermore, replacement of structurally similar amino acids between the CTD of CpxII and the SNAP25-SN1 domain on the polar side of the amphipathic helix had no functional consequences (*Figure 2—figure supplement 2*). Thus, an unperturbed hydrophobic face of the amphipathic helix is essential for the fusion-inhibiting effect of the CpxII CTD. In particular, experiments on the *Caenorhabditis elegans* NMJ suggested that membrane binding of the CpxII CTD via its amphipathic helix sequesters the protein to synaptic vesicles, thereby, concentrating other SNARE-binding regions of the protein (e.g. the accessory helix at the site of exocytosis for efficient molecular clamping; *Wragg et al., 2013*; *Snead et al., 2014*; *Gong et al., 2016*). However, other structure-function analyses with Cpx from *C. elegans* have shown that membrane binding is important but does not suffice for the inhibitory effect of Cpx (*Snead et al., 2017*; *Wragg et al., 2017*). Our biochemical pull-down experiments first show that the isolated CTD of CpxII binds to SNARE proteins (i.e. syntaxin, SybII) and SytI (*Figure 3*). Second, they demonstrate that the differential restoration of exocytosis by CpxII and its mutant variants mirrors its altered binding to SNAREs and SytI. Furthermore, compared to the WT protein, none of these CTD mutants exhibited defects in binding to vesicular membranes (*Figure 3—figure supplement 2*). Thus, mutants with unaltered membrane binding showed a profound deficit in the prevention of premature exocytosis and the formation of a primed vesicle pool rendering the presumed necessary concentration of CpxII on secretory organelles as a genuine mechanism for the inhibitory action of the CTD unlikely. Evidently, protein-protein rather than protein-membrane interactions of the CpxII CTD determine the mechanism of molecular clamping in chromaffin cells. Nevertheless, our findings confirm membrane binding by the CpxII CTD and do not exclude additional membrane sculpting activity of Cpx, as observed by nanodisc-black lipid membrane electrophysiology (*Courtney et al., 2022*). As Cpx binds with much higher affinity to binary acceptor complexes than to membranes (*Zdanowicz et al., 2017*), the Cpx CTD may switch from a membrane-bound to a protein-bound state in the course of vesicle docking to the plasma membrane. Furthermore, an unaltered fusion-inhibitory phenotype of the CpxII-CTD can be observed in the absence of SytI (*Figure 5—figure supplement 1*) or SytVII (*Figure 6*) and in the joint absence of both Ca$^{2+}$ sensors in chromaffin cells (*Dhara et al., 2014*) which counters alternative mechanisms such as a functional antagonism of the Ca$^{2+}$ sensor SytI or of secondary Ca$^{2+}$ sensors like SytVII in chromaffin cells (*Yang et al., 2010*). Indeed, recent structural analyses of the formation of individual SNARE complexes using optical tweezers confirm that the CTD of CpxII plays an active role in the arrest of assembling SNARE complexes (*Hao et al., 2023*). We have previously shown that acute infusion of the isolated C-terminal peptide of CpxII into WT cells significantly diminishes premature vesicle fusion, but fails to clamp tonic secretion in CpxII ko cells, indicating that other domains of the Cpx protein are required to cooperate with the CTD (*Makke et al., 2018*). As the accessory helix of Cpx has been found to bind to membrane-proximal cytoplasmic regions of SNAP-25 and SybII (*Malsam et al., 2012*; *Bykhovskaia et al., 2013*; *Vasin et al., 2016*; *Bera et al., 2022*), an attractive scenario could be that both domains of CpxII, the CTD and the accessory helix, synergistically cooperate to stall final SNARE assembly. In summary, our studies provide new functional insights into fundamental mechanisms of secretion control and support a model wherein the CpxII CTD, with the hydrophobic face of its amphipathic helix, prevents untimely fusion by arresting the progressive assembly of membrane-bridging SNARE complexes prior to the actual Ca$^{2+}$ stimulus.

## Lifting the clamp

Locking of fusion-promoting SNARE complexes requires a fast-acting mechanism to release the clamp in order to meet the speed requirements of rapid Ca$^{2+}$-driven exocytosis. While recent models propose a mechanism by which concurrent binding of SytI and Cpx to partially assembled SNARE complexes

defines the primed vesicle state, no consensus has been reached, despite tremendous efforts, on the precise mechanism by which $Ca^{2+}$ binding to the C2-domains of SytI triggers stimulus-secretion coupling (for review, see *Brunger et al., 2019*; *Rizo, 2022*). Previous biochemical experiments have shown that SytI interacts with Cpx by binding to a polyacidic glutamate cluster immediately upstream of the amphipathic helix (*Tokumaru et al., 2008*). Even though the long acidic stretch is conserved across the animal kingdom (*Lottermoser and Dittman, 2023*), no functional significance has been ascribed to it. Our observations show that substitution of the glutamate residues with a corresponding stretch of alanine residues reduced the EB size and most remarkably slowed down exocytosis timing (*Figure 4*). The latter result indicates that Cpx's CTD cooperates with SytI in stimulus-secretion coupling and suggests that SytI interaction with the polyacidic glutamate cluster is instrumental in lifting the clamp imposed by Cpx's amphipathic helix. Notably, a comparable slowing in exocytosis timing was not observed in the absence of the entire CTD of CpxII (i.e. by deleting the amino acid positions 101–134, *Makke et al., 2018*), suggesting that this SytI-CpxII interaction site is specifically required for lifting the clamp signal imposed by Cpx's downstream amphipathic helix. While CpxII aligns with its central helix in an antiparallel orientation on the SNARE complex, downstream protein regions of CpxII may provide sufficient structural flexibility for the far C-terminus to fold back in a parallel orientation on membrane-proximal layers of the partially assembled SNARE complex (*Bowen et al., 2005*; *Makke et al., 2018*; *Hao et al., 2023*). Under these conditions, the glutamate cluster of Cpx's CTD may come into close proximity to the position of crucial amino acids of SNAP-25 (i.e. D51, E52, E55), which together with R398, R399, and neighboring basic residues (K288, R281) in the C2B of SytI form the SytI-SNARE primary interface (*Zhou et al., 2015*; *Wang et al., 2016*). The primary interface is critical for SytI's function to efficiently trigger release in cultured neurons and chromaffin cells (*Gaffaney et al., 2008*; *Mohrmann et al., 2013*; *Rickman et al., 2006*; *Schupp et al., 2016*; *Xue et al., 2007*; *Zhou et al., 2015*). Indeed, mutations of amino acids of SNAP25 SN2 (such as E55, D51, E52) that line the SytI-SNARE primary interface resulted in a phenotype similar to that of the CpxII glutamate cluster mutation (*Mohrmann et al., 2013*). Furthermore, mutations within SNAP25 (e.g. D166A, E170E, *Mohrmann et al., 2013*), which contribute to the primary SytI-SNARE interface and are located on the opposite side of the Cpx-bound SNARE complex (*Chen et al., 2002*), impair Cpx-mediated clamping of liposome fusion in vitro (*Schupp et al., 2016*), an observation consistent with the hypothesis of an independent binding site of the CpxII CTD on the SNARE complex. Collectively, these results point to a structurally plausible co-localization of 'triggering' and 'unclamping' mechanisms on the surface of the SNARE complex.

## SytI-CpxII interplay

By analyzing the impact of single and combined deficiencies for CpxII and SytI (*Cplx2^-/-^; Syt1^-/-^; Cplx2^-/-^:Syt1^-/-^* dko), we found that CpxII functions interdependently with SytI (*Figure 5*). Additional loss of SytI in the absence of CpxII had no impact on either the magnitude or the kinetics of exocytosis. This suggests that all functions of SytI depend on CpxII. In sharp contrast, additional loss of SytVII in the absence of CpxII further aggravated the secretion deficit (*Cplx2^-/-^:Syt1^-/-^* dko, *Figure 5*). Moreover, CpxII expression experiments show that CpxII fails to boost exocytosis in the absence of SytI, but is able to do so in the absence of SytVII (*Figure 5—figure supplement 2*). Remarkably, the profound loss of secretion in the *Cplx2^-/-^:Syt7^-/-^* double deficiency closely resembles the phenotype of the *Syt1^-/-^:Syt7^-/-^* deficiency (*Schonn et al., 2008*; *Dhara et al., 2014*), indicating the CpxII serves as a gatekeeper for all SytI functions. In murine hippocampal neurons, loss of *Cplx1* and *Syt1* has additive effects on fast synchronous release, suggesting independent mechanisms (*Xue et al., 2010*). On the other hand, the same study also showed that *Syt1* heterozygosity fails to reduce release probability in WT neurons, but does so in the absence of Cpx, again suggesting that Cpx and Syt1 may functionally interact in $Ca^{2+}$-triggered release. Recent structural evidence provides potential cues explaining the differential impact of SytI or SytVII deficiencies in the absence of CpxII (*Zhou et al., 2015*; *Zhou et al., 2017*). For instance, core residues in SytI that are essential for SNARE binding at the 'primary' interface are not conserved in SytVII, which has been implicated as $Ca^{2+}$ sensor for asynchronous synaptic vesicle exocytosis (*Wen et al., 2010*; *Bacaj et al., 2013*; *Liu et al., 2014*; *Jackman et al., 2016*). Thus, it seems unlikely that SytVII engages the SNARE complex in the same manner as SytI. SytI and Cpx, instead, form a separated but continuous α-helix at the 'tripartite' SytI-Cpx-SNARE interface, which may serve as pre-fusion intermediate for the primed vesicle state (*Zhou et al., 2017*). Furthermore,

sequence alignment reveals that amino acids involved in specific side-chain interactions of the tripartite interface are only highly conserved in SytI, SytII, and SytIX and thus in the isoforms involved in rapid evoked release. Taken together, our results show that SytI and CpxII acts as interdependent allies in rapid secretion, whereas SytVII acts autonomously and thereby additively as $Ca^{2+}$ sensor that accelerates vesicle recruitment during train stimulation, resulting in sustained release (*Liu et al., 2014*; *Tawfik et al., 2021*).

Despite extensive research, there is still no consensus on the mechanisms underlying the fusion-promoting function of Cpx. Using hippocampal microisland cultures, it was found that the NTD of CpxI (amino acid positions 1–26) is required to restore basal evoked transmitter release (*Xue et al., 2007*) and that the NTD, by binding to the C-terminal end of the SNARE complex, provides conformational support to the SNARE machinery (*Xue et al., 2010*). In the same line, individual liposome-liposome content mixing experiments with the minimal fusion machinery showed that the Cpx-N terminus decisively improves the fusion fidelity (*Lai et al., 2014*). On the other hand, Cpx deficiency was found to be associated with a reduction in the $Ca^{2+}$ sensitivity of evoked release in various preparations, including neurons and endocrine cells (*Reim et al., 2001*; *Tadokoro et al., 2005*; *Huntwork and Littleton, 2007*; *Xue et al., 2007*; *Jorquera et al., 2012*; *Cho et al., 2014*; *Dhara et al., 2014*). In the present work, we show that the CpxII-NTD accelerates synchronous exocytosis by enhancing its apparent $Ca^{2+}$ affinity in a SytI-dependent manner (*Figures 7 and 8*). While an excess of CpxII rescues the slow release kinetics of the SytI R233Q mutant (*Sørensen et al., 2003*), this is further slowed by the expression of the CpxII ΔN mutant (*Figure 7*). In contrast, in the absence of SytI, no corresponding changes were observed with expression of the CpxII ΔN mutant (*Figure 6*). The latter result confirms the strongly interdependent action of CpxII and SytI and is difficult to reconcile with a potential direct action of the CpxII NTD on SNARE assembly (*Xue et al., 2010*). Moreover, in response to slow ramp-like $Ca^{2+}$ stimuli, CpxII and its ΔN mutant led to opposite shifts in the $Ca^{2+}$ dependence of secretion (*Figure 8C*) and to corresponding changes in the fusion rate constant as the function of [Ca]i (*Figure 8D*). The results favor a model wherein the CpxII NTD either directly regulates the biophysical properties of the $Ca^{2+}$ sensor by increasing the forward rate of $Ca^{2+}$ binding or indirectly affects SytI-SNARE or SytI-membrane interactions, thereby, lowering the energy barrier of $Ca^{2+}$-triggered fusion.

Taken together, Cpx impedes SNARE assembly via the hydrophobic face of the amphipathic helix at its C-terminus and simultaneously provides structures with the upstream glutamate cluster that adjust SytI-dependent triggering of the fusion mechanism. Complexin thus maintains a delicate balance between preventing premature release and triggering rapid exocytosis by manipulating the assembly of SNARE motifs and assisting SytI in triggering efficient rapid exocytosis.

# Materials and methods

## Key resources table

| Reagent type (species) or resource | Designation | Source or reference | Identifiers | Additional information |
|---|---|---|---|---|
| Strain, strain background (*Mus musculus*) | C57BL/6 | Jackson Lab strain: # 000664 | RRID:IMSR_JAX:000664 | |
| Genetic reagent (*Mus musculus*) | *Cplx 2* [-/-] | *Reim et al., 2001* | PMID:11163241 | |
| Genetic reagent (*Mus musculus*) | *Syt 1* [-/-] | Jackson Lab strain: # 002478 | PMID:7954835 | |
| Genetic reagent (*Mus musculus*) | *Syt 7* [-/-] | Jackson Lab Strain: # 004950 | PMID:18308933 | |
| Genetic reagent (*Mus musculus*) | *Syt 1* [-/-] /*Syt 7* [-/-] | This paper: Syt1: Jackson Lab Strain: # 002478 Syt7: Jackson Lab Strain: # 004950 | PMID:18308932 | |

*Continued on next page*

*Continued*

| Reagent type (species) or resource | Designation | Source or reference | Identifiers | Additional information |
|---|---|---|---|---|
| Genetic reagent (*Mus musculus*) | Syt 1 R233Q ki | *Fernández-Chacón et al., 2001*; The Jackson laboratory B6;129P2-Syt1tm3Sud/J RRID:IMSR_JAX:006385 | PMID:11242035 | |
| Antibody | Mouse monoclonal anti-Syntaxin1 | Synaptic Systems | Cat# 110 001 | Western blot: 1:1000 |
| Antibody | Mouse monoclonal anti-SNAP25 | Synaptic Systems | Cat# 111 011 | Western blot: 1:1000 |
| Antibody | Mouse monoclonal anti-SynaptobrevinII | Synaptic Systems | Cat# 104 211 | ICC, western blot: 1:1000 |
| Antibody | Mouse monoclonal anti-Synaptotagmin I | Synaptic Systems | Cat# 105 011 | Western blot: 1:1000 |
| Antibody | Mouse monoclonal anti-Synapsin 1 | Synaptic Systems | Cat# 106 001 | Western blot: 1:1000 |
| Antibody | Rabbit polyclonal anti-CpxII | This paper | Materials and methods | ICC, western blot 1:5000 |
| Antibody | HRP conjugated goat-anti mouse | Bio-Rad Laboratories | Cat# 170-5047 | Western blot: 1:1000 |
| Antibody | HRP conjugated goat-anti rabbit | Bio-Rad Laboratories | Cat# 170-5046 | Western blot: 1:1000 |
| Antibody | Alexa Fluor 555 goat anti-mouse | Invitrogen | Cat# A21422 | ICC: 1:1000 |
| Antibody | Alexa Fluor 488 goat anti-rabbit | Invitrogen | Cat# A11008 | ICC: 1:1000 |
| Recombinant DNA reagent (*Mus musculus*) | *Cplx2*-WT | GenBank: U35101.1 | | |
| Transfected construct (*Mus musculus*) | pSFV-CpxII 27-134 (CpxII ΔN) IRES-EGFP | This paper | | Derived from U35101.1 with indicated mutations, Semliki Forest virus expression construct, see Mutagenesis and viral constructs |
| Transfected construct (*Mus musculus*) | pSFV-CpxII 1-115-IRES-EGFP | This paper | | Derived from U35101.1 with indicated mutations, Semliki Forest virus expression construct, see Mutagenesis and viral constructs |
| Transfected construct (*Mus musculus*) | pSFV-CpxII 1-115-amphepathic helix IRES-EGFP | This paper | | Derived from U35101.1 with indicated mutations, Semliki Forest virus expression construct, see Mutagenesis and viral constructs |
| Transfected construct (*Mus musculus*) | pSFV-CpxII L124W-L128W IRES-EGFP | This paper | | Derived from U35101.1 with indicated mutations, Semliki Forest virus expression construct, see Mutagenesis and viral constructs |
| Transfected construct (*Mus musculus*) | pSFV-CpxII L124E-L128E IRES-EGFP | This paper | | Derived from U35101.1 with indicated mutations, Semliki Forest virus expression construct, see Mutagenesis and viral constructs |
| Transfected construct (*Mus musculus*) | pSFV-CpxII L117W-L121W IRES-EGFP | This paper | | Derived from U35101.1 with indicated mutations, Semliki Forest virus expression construct, see Mutagenesis and viral constructs |

*Continued*

| Reagent type (species) or resource | Designation | Source or reference | Identifiers | Additional information |
|---|---|---|---|---|
| Transfected construct (*Mus musculus*) | pSFV-CpxII L124E-L128E IRES-EGFP | This paper | | Derived from U35101.1 with indicated mutations, Semliki Forest virus expression construct, see Mutagenesis and viral constructs |
| Transfected construct (*Mus musculus*) | pSFV-CpxII Q129A IRES-EGFP | This paper | | Derived from U35101.1 with indicated mutations, Semliki Forest virus expression construct, see Mutagenesis and viral constructs |
| Transfected construct (*Mus musculus*) | pSFV-CpxII D118A IRES-EGFP | This paper | | Derived from U35101.1 with indicated mutations, Semliki Forest virus expression construct, see Mutagenesis and viral constructs |
| Transfected construct (*Mus musculus*) | pSFV-CpxII D130A IRES-EGFP | This paper | | Derived from U35101.1 with indicated mutations, Semliki Forest virus expression construct, see Mutagenesis and viral constructs |
| Transfected construct (*Mus musculus*) | pSFV-CpxII K133A IRES-EGFP | This paper | | Derived from U35101.1 with indicated mutations, Semliki Forest virus expression construct, see Mutagenesis and viral constructs |
| Transfected construct (*Mus musculus*) | pSFV-CpxII D118K-D130K IRES-EGFP | This paper | | Derived from U35101.1 with indicated mutations, Semliki Forest virus expression construct, see Mutagenesis and viral constructs |
| Transfected construct (*Mus musculus*) | pSFV-CpxII E-A IRES-EGFP | This paper | | Derived from U35101.1 with indicated mutations, Semliki Forest virus expression construct, see Mutagenesis and viral constructs |
| Recombinant DNA reagent | pGEX-KG-vector | This paper | | Prokaryotic expression vector, see Biochemistry |
| Peptide, recombinant protein | GST-C-terminal domain peptide | This paper | | Materials and methods |
| Peptide, recombinant protein | GST-C-terminal domain peptide L124E-L128E | This paper | | Materials and methods |
| Peptide, recombinant protein | GST-C-terminal domain peptide L124W-L128W | This paper | | Materials and methods |
| Peptide, recombinant protein | GST-C-terminal domain peptide amphipathic helix | This paper | | Materials and methods |
| Software algorithm | IgorPro | WaveMetrics Software | | |
| Software algorithm | AutesP | NPI electronics | | |
| Software algorithm | Zen2008 | Zeiss | | |
| Software algorithm | ImageJ | National Institutes of Health | | |

## Mutagenesis and viral constructs

Briefly, mutations in CpxII were generated by overlap extension polymerase chain reaction. For expression in chromaffin cells, cDNAs encoding for CpxII or its mutant variants were sub-cloned into the first open reading frame (ORF) of a bicistronic Semliki Forest vector (pSFV1, Invitrogen, San Diego, CA, USA). Enhanced-GFP expression from the second ORF allowed for the identification of infected cells.

## Culture of chromaffin cells and electrophysiological recordings

All experiments were performed on mouse chromaffin cells prepared at postnatal day 0–1 from ko/ki pups and their littermate control (WT or heterozygous) which were identified by genotyping. Preparation of adrenal chromaffin cells was done as described previously (*Borisovska et al., 2005*).

Electrophysiological recordings were carried out at DIV2, 5.5 hr after addition of viral particles to the cultured cells.

Electrophysiological recordings were done at room temperature. Cells were recorded in an extracellular Ringer's solution containing (in mM): 130 NaCl, 4 KCl, 2 CaCl$_2$, 1 MgCl$_2$, 30 glucose, 10 HEPES-NaOH, pH 7.3, 310 mOsm. Recordings of membrane capacitance (reflecting vesicle fusion) and ratiometric [Ca$^{2+}$]i changes (using fura-2 and furaptra) were performed as described previously (*Borisovska et al., 2005*). The intracellular solution for Ca$^{2+}$-uncaging experiments contained (in mM): 110 Cs-glutamate, 8 NaCl, 3.5 CaCl$_2$, 5 NP-EGTA, 0.2 fura-2, 0.3 furaptra, 2 MgATP, 0.3 Na$_2$GTP, 40 HEPES-CsOH, pH 7.3, 300 mOsm. Membrane capacitance was recorded with the Pulse software (HEKA, Lambrecht, Germany) and CM were performed according to the Lindau-Neher technique (sine wave stimulus: 1000 Hz, 35 mV peak-to-peak amplitude, DC-holding potential –70 mV). Synchronized vesicle exocytosis was stimulated by a brief UV-flash that led to Ca$^{2+}$ uncaging upon photolysis of NP-EGTA. Priming of secretory vesicles was promoted by infusing cells for 2 min with the intracellular solution containing 500 nM free [Ca$^{2+}$]i. During this phase premature vesicle secretion was assessed. The flash-evoked capacitance increase was approximated with the function: $f(x) = A0 + A1(1 - \exp[-t/\tau 1]) + A2(1 - \exp[-t/\tau 2]) + kt$ (*Rettig and Neher, 2002*). For calcium ramp experiments, after vesicle priming, intracellular Ca$^{2+}$ was gradually uncaged by the continuous UV illumination from the monochromator. The resulting membrane capacitance increase was approximated with a polynomial function. The size of the synchronized EB, the remaining pool, and the slope of the capacitance trace were determined at fixed time intervals of 40 ms ($\Delta$CM/0.04 s) for each cell to match the time interval of the Ca$^{2+}$ measurement. The secretion rate for each interval was determined by rate [1/s] = slope [fF/s]/ remaining pool [fF]. For averaging data from different cells, the corresponding rates were plotted against the Ca$^{2+}$ rise interpolated at 50 nM Ca$^{2+}$ intervals. All signals were analyzed with customized IgorPro routines (WaveMetrics, Lake Oswego, OR, USA).

## Immunocytochemistry

Chromaffin cells were processed either 5.5 hr (*Figure 3—figure supplement 1*) or for 2.5 hr (*Figure 3—figure supplement 2*) after virus infection as described previously (*Makke et al., 2018*). A homemade, affinity-purified rabbit polyclonal antibody against CpxII (epitope: amino acids 1–100 of CpxII) was used for all immunofluorescence experiments described in the manuscript. In co-localization experiments (*Figure 3—figure supplement 2*), chromaffin cells were co-stained with rabbit polyclonal CpxII and mouse monoclonal SybII antibodies (clone 69.1, antigen epitope amino acid position 1–14, kindly provided by R Jahn, MPI for Biophysical Chemistry, Göttingen, Germany). Immunopositive signals were determined after threshold adjustment (4× background signal) and cytofluorgram as well as Pearson's co-localization coefficient were analyzed with ImageJ (JACoP plugin).

## Biochemistry

Recombinant N-terminal tagged GST fusion proteins (pGEX-KG-vector with an internal thrombin cleavage site) were expressed in the *Escherichia coli* strain BL21DE3 and purified using glutathione-agarose according to the manufacturer's instructions. 250 µg recombinant GST-CpxII-CTD or its mutant variants were incubated with 100 µl GST bead slurry for 1 hr at room temperature while shaking at 1000 rpm (Vibrax, IKA, Staufen, Germany). Beads were then centrifuged (200 rpm, 3 min) and washed three times with assay buffer containing: 130 mM NaCl, 50 mM HEPES, 1 mM EDTA, 1 mM DTT, 1% Triton X-100, and complete protease inhibitor. Integrity and purity of bound GST protein was verified by Coomassie staining. 170 µl Triton X-100 extract of mouse brain homogenate (0.7 mg/ml, containing 130 mM NaCl, 50 mM HEPES-NaOH, 1 mM EDTA, 2% Triton X-100, 1 mM PMSF, pH 7.3) was incubated with 50 µl beads (2 hr, room temperature, final assay volume 250 µl). Immobilized proteins were eluted by thrombin cleavage (thrombin 0.02 UN/µl, Sigma, Germany). The unbound fraction (*Sbh*), the thrombin eluted fraction (*STh*), and the pellet fraction after thrombin cleavage (*P*) were analyzed by 12% SDS-PAGE and western blotting (*Figure 3*). For the detection of Syntaxin 1A, SNAP25, SybII, SytI, and Synapsin1 the following mouse antibodies from Synaptic Systems (Göttingen, Germany) were used: Anti-Syntaxin1 (CL 78.2) No: 110 001; SNAP25 (CL 71.1), No:111 011; SynaptobrevinII (CL 69.1), No: 104 211; Synaptotagmin 1 (CL 41.1), No:105 011; Anti-Synapsin 1 (CL 46.1), No:106 001. Primary mouse antibodies were used at a dilution of 1:1000. Immunoreactive bands were visualized with secondary goat-anti mouse or goat anti-rabbit antibodies conjugated with horseradish

peroxidase and with an enhanced chemiluminescence system (Thermo Fisher Scientific, Schwerte, Germany).

## Statistical analysis

Data in bar graphs and X-Y plots present means ± SEM. Statistical analyses were performed using Prism 7 (GraphPad). All statistical data is summarized in the corresponding 'Source Data' tables and were tested for normality with Kolmogorov-Smirnov test. Data from groups with nonparametric distribution was subjected to Kruskal-Wallis followed by Dunn's post hoc test when at least one group showed a nonparametric distribution. Data from groups showing parametric distributions was subjected to ordinary one-way ANOVA followed by Tukey-Kramer post hoc test. Significance levels: '*' $p<0.05$, '**' $p<0.01$, and '***' $p<0.001$.

## Acknowledgements

We thank Marina Wirth. This work was supported by the Collaborative Research Center 1027 to DB.

## Additional information

### Funding

| Funder | Grant reference number | Author |
| --- | --- | --- |
| Deutsche Forschungsgemeinschaft | CRC 1027 | Dieter Bruns |

The funders had no role in study design, data collection and interpretation, or the decision to submit the work for publication.

### Author contributions

Mazen Makke, Data curation, Formal analysis, Writing – original draft; Alejandro Pastor-Ruiz, Antonio Yarzagaray, Surya Gaya, Data curation, Formal analysis; Michelle Zimmer, Walentina Frisch, Data curation; Dieter Bruns, Conceptualization, Data curation, Writing – original draft, Writing – review and editing

### Author ORCIDs

Surya Gaya ⬤ https://orcid.org/0000-0003-0163-5748
Dieter Bruns ⬤ https://orcid.org/0000-0002-2497-1878

### Ethics

All experimental procedures were approved by and performed according to the welfare regulations and ethical guidelines from the local governing body (Amtstierärztlicher Dienst, Saarland Germany, approval number: 2.4.1.1-CIPMM and GB3-2.4.7.1).

Reviewer #1 (Public Review): https://doi.org/10.7554/eLife.92438.4.sa1
Reviewer #2 (Public Review): https://doi.org/10.7554/eLife.92438.4.sa2
Author response https://doi.org/10.7554/eLife.92438.4.sa3

## Additional files

### Supplementary files
• MDAR checklist

### Data availability

This study includes no data deposited in external repositories. All relevant data are included in the article and/or its expanded figures, and the source data supplementary information files. Information and requests for resources and reagents should be directed to the corresponding author.

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
