## [Editor Report · eLife assessment]

This **important** work shows **compelling** data that significantly advances our understanding of the regulation of neurotransmitter and hormone secretion by exploring the mechanisms of how the protein complexin 2 (Cplx2) interacts with the calcium sensor synaptotagmin. The function of mammalian Cplx2 is studied using chromaffin cells derived from Cplx2 knock out mice as a system to overexpress and functionally characterize mutant Cplx2 forms and the interaction between Cplx2 and synaptotagmin. The authors identify structural requirements within the protein for Cplx's dual role in preventing premature vesicle exocytosis and enhancing evoked exocytosis. The findings are of broad interest to neuroscientists and cell biologists.

---

## [Referee Report · Reviewer #1 (Public Review)]

Summary:

Using chromaffin cells as a powerful model system for studying secretion, the authors study the regulatory role of complexin in secretion. Complexin is still enigmatic in its regulatory role, as it both provides inhibitory and facilitatory functions in release. The authors perform an extensive structure-function analysis of both the C- and N-terminal regions of complexin. There are several interesting findings that significantly advances our understanding of cpx/SNARe interactions in regulating release. C-terminal amphipathic helix interferes with SNARE complex assembly and thus clamps fusion. There are acidic residues in the C-term that may be seen as putative interaction partners for Synaptotagmin. The N-terminus of Complexin promoting role may be associated with an interaction with Syt1. In particular the putative interaction with Syt1 is of high interest and supported by quite strong functional and biochemical evidence. The experimental approaches are state of the art, and the results are of the highest quality and convincing throughout. They are adequate and intelligently discussed in the rich context of the standing literature. Whilst there are some concerns about whether the facilitatory actions of complexion have to be tightly linked to Syt1 interactions, the proposed model will significantly advance the field by providing new directions in future research.

---

## [Referee Report · Reviewer #2 (Public Review)]

Summary:

Complexin (Cplx) is expressed at nearly all chemical synapses. Mammalian Cplx comes in four different paralogs which are differentially expressed in different neurons or secretory cell types, either selectively or in combination with one or two other Cplx isoforms. Cplx binds with high affinity to assembled SNARE complexes and promotes evoked synchronous release. Cplx is assumed to preclude premature SV fusion by preventing full SNARE assembly, thereby arresting subsequent SNARE-driven fusion ("fusion-clamp" theory). The protein has multiple domains, the functions of which are controversially discussed. Cplx's function has been studied in a variety of model organisms including mouse, fly, worm, and fish with seemingly conflicting results which led to partly contradicting conclusions.

Makee et al. study the function of mammalian Cplx2 in chromaffin cells by making use of Cplx2 ko mice to overexpress and functionally characterize mutant Cplx2 forms in cultured chromaffin cells. The main conclusion of the present study are:

The hydrophobic character of the amphipathic helix in Cplx's C-terminal domain is essential for inhibiting premature vesicle fusion at a [Ca2+]i of several hundreds of nM (pre-flash [Ca2+]i). The Cplx-mediated inhibition of fusion under these conditions does not rely on expression of either Syt1 or Syt7.

Slow-down of exocytosis by N-terminally truncated Cplx mutants in response to a [Ca2+]i of several µM (peak flash [Ca2+]i) occurs regardless of the presence or absence of Syt7 demonstrating that Cplx2 does not act as a switch favoring preferential assembly of the release machinery with Syt1,2 rather than the "slow" sensor Syt7.

Cplx's N-terminal domain is required for the Cplx2-mediated increase in the speed of exocytosis and faster onset of exocytosis which likely reflect an increased apparent Ca2+ sensitivity and faster Ca2+ binding of the release machinery.

Strengths:

The authors perform systematic truncation/mutational analyses of Cplx2. They analyze the impact of single and combined deficiencies for Cplx2 and Syt1 to establish interactions of both proteins.

State-of-the-art methods are employed: Vesicle exocytosis is assayed directly and with high resolution using capacitance measurements. Intracellular [Ca2+] is controlled by loading via the patch-pipette and by UV-light induced flash-photolysis of caged [Ca2+]. The achieved [Ca2+] is measured with Ca2+ -sensitive dyes.

The data is of high quality and the results are compelling.

Weaknesses:

With the exception of mammalian retinal ribbon synapses (and some earlier RNAi knock down studies which had off-target effects), there is little experimental evidence for a "fusion-clamp"-like function of Cplxs at mammalian synapses. At conventional mammalian synapses, genetic loss of Cplx (i.e. KO) consistently decreases AP-evoked release, and generally either also decreases spontaneous release rates or does not affect spontaneous release, which is inconsistent with a "fusion-clamp" theory. This is in stark contrast to invertebrate (D. m. and C. e.) synapses where genetic Cplx loss is generally associated with a strong upregulation of spontaneous release.

There are alternative scenarios explaining how Cplx may phenomenological "clamp" vesicle fusion rates without mechanistically assigning a "clamping" function to Cplx (Neher 2010, Neuron). In fact, changes in asynchronous release kinetics following conditioning AP trains observed at Cplx1 ko calyx of Held synapses do not favor a "fusion clamp" model (Chang et al., 2015, J.Neurosci.), while an alternative model, assigning Cplx the role of a "checkpoint" protein in SNARE assembly, quantitatively reproduces all experimental observations (Lopez et al., 2024, PNAS). It might be helpful for a reader to mention such alternative scenarios.

---

## [Author Response]

The following is the authors’ response to the previous reviews.

**Reviewer #1 (Recommendations For The Authors):**
The revised manuscript addressed my minor concerns adequately, and the manuscript is now further improved. I have no remaining criticisms.
**Reviewer #2 (Recommendations For The Authors):**
Abstract:line 45 The abbreviation "SytI" should perhaps be introduced above.

done

Results:line 139 "RRP kinetics" should perhaps read "RRP depletion kinetics" or "secretion kinetics".

We replaced “RRP kinetics” with “RRP secretion kinetics”

line 325ff and Figure 8As far as I understand, SytI 875 R233Q ki cells shown in violet express wt CplxII. Perhaps this should be explicitly stated?

To accommodate this suggestion: We now state on page 13 line 302: “Overexpression of the CpxII DN mutant in SytI R233Q ki cells, which is expected to outcompete the function of endogenous CpxII in these cells (Dhara et al., 2014), further slowed down the rate of synchronized release and restored the EB size to the wt level (Figure 7C, D)”

line 332ff and Figure 8What is plotted in Figure 8B bottom and in Figure 8D is not a "rate" but rather a "unitary rate", more commonly referred to as a "rate constant".The y-axis label of Figures 8B and 8D should therefore better be changed to "rate constant". See also line 528 of the Discussion.

Figure (y-axis label) and text were changed accordingly